# How automatic speed control based on distance affects user behaviours in telepresence robot navigation within dense conference-like environments

**Anil Ufuk Batmaz**[1]*, **Jens Maiero**[2], **Ernst Kruijff**[2], **Bernhard E. Riecke**[1], **Carman Neustaedter**[1], **Wolfgang Stuerzlinger**[1]

**1** School of Interactive Arts & Technology, Simon Fraser University, Vancouver, BC, Canada, **2** Institute of Visual Computing Bonn-Rhein-Sieg University of Applied Sciences, Sankt Augustin, Germany

* abatmaz@sfu.ca

**Data Availability Statement:** All anonymous user study files are available from the Open Science Framework database (osf.io/anxv3/).

## Abstract

Telepresence robots allow users to be spatially and socially present in remote environments. Yet, it can be challenging to remotely operate telepresence robots, especially in dense environments such as academic conferences or workplaces. In this paper, we primarily focus on the effect that a speed control method, which automatically slows the telepresence robot down when getting closer to obstacles, has on user behaviors. In our first user study, participants drove the robot through a static obstacle course with narrow sections. Results indicate that the automatic speed control method significantly decreases the number of collisions. For the second study we designed a more naturalistic, conference-like experimental environment with tasks that require social interaction, and collected subjective responses from the participants when they were asked to navigate through the environment. While about half of the participants preferred automatic speed control because it allowed for smoother and safer navigation, others did not want to be influenced by an automatic mechanism. Overall, the results suggest that automatic speed control simplifies the user interface for telepresence robots in static dense environments, but should be considered as optionally available, especially in situations involving social interactions.

## Introduction

Telepresence robots (TRs) are designed to allow a remote user to have a mobile presence in a remote physical space. They consist of a video conferencing display and series of cameras that are attached to a robotic 'body' of some form with wheels to support moving and remote driving. TRs have been used and tested in various scenarios, including, but not limited to, working in remote office settings [1–5], schooling for home-bound children [6, 7] and shopping [8]. All of these studies have shown that TRs can increase social presence in remote environments [5, 8, 9]. However, using a TR in a remote environment is not always easy. Navigation challenges can arise due to the limited field of view of the TR's camera [1], the limited resolution of the

**Funding:** The author(s) received no specific
funding for this work.

**Competing interests:** The authors have declared
that no competing interests exist.

camera images [1], the need to avoid both obstacles and people [2], and potentially non-intuitive input methods for controlling the robot [8]. Many of these problems may result in a limited situational awareness of the user. Yet, such awareness is normally required to effectively navigate around. As a result of these challenges, researchers have called for additional control mechanisms or forms of feedback to aid TR drivers, such as wider fields of view [10, 11] and additional cameras [1].

Inspired by work in Virtual Reality navigation, e.g., [12, 13], and in the automotive sector, e.g., [14, 15], we implemented a *speed control* (SC) method that uses sensory data from ultrasonic range finders to automatically slow down the TR as it gets closer to potential obstacles it might otherwise bump into, typically because the remote user misjudged the speed and/or distance. Thus, the potential need for a support system, such as SC, i.e., functionality that controls how fast the robot moves forward, is higher when the user has to drive through a dense environment with narrow passages, such as office environments with narrow corridors, a gathering of many people in a conference hall, or other forms of social gatherings, where TR users interact with other people. Previous work on speed control in industrial robotics had been reported in the literature [16–18]. Recently, SC methods have been introduced to tele-robotics systems. However, to our knowledge, the use of distance-based speed control algorithms has not been studied for TRs in dense environments nor with tasks in a conference-like environment. This motivated us to evaluate distance-based SC in such environments and assess the change in navigation behaviours of users.

Our work builds on perceptual load theory: humans have limited attentional resources, and as users driving TRs often exhibit higher cognitive load [17], we assume that using an automatic SC algorithm for a TR could help people to free attention and reduce cognitive load. This would reduce collisions and improve navigation behaviours, spatial awareness, and presence. Hence, instead of taking over navigation, the system is designed to support user interaction. We do so by adding a simple distance-based SC algorithm to the TR platform, the (original) Beam+ by Suitable Technologies.

Based on this hardware platform, we present two major contributions. Our first contribution is the evaluation of user navigation behaviours with a TR with distance-based SC in dense environments. We evaluated these behaviours in our first user study and showed that distance-based SC can improve navigation behaviour in terms of reducing the number of collisions while not increasing task execution time. Our second contribution concerns social interaction with SC assistance: TRs are designed to enable social interaction with remote people. Thus, we designed a second study where participants had to not only navigate in a dense environment but also socially interact with people. While SC slightly reduced task completion time and number of collisions here, participant preferences on the SC changed when they interacted socially.

These results suggest that SC algorithms may not improve TR navigation behaviour in every case, especially in a social environment. Overall, our results show that SC significantly improves TR navigation behaviours through a reduction in the number of collisions in static dense environments, but not necessarily when interacting with people. These findings suggest that methods for automatically adjusting TR speed based on proximity to objects are promising, however, design work needs to carefully consider that activation of SC should still be controlled by the user.

## 1 Previous work

The design and usage of TRs has been widely studied. Researchers have found that TRs provide stronger feelings of presence in a variety of remote environments (e.g., conferences,

schools, hospitals) when compared to using video conferencing systems, due to one's ability to move around and have a physical body in the remote space [1, 2, 5–7]. As a result, TRs have been shown to enhance relationships, including connections between co-workers [5], school friends [6], and long-distance romantic partners [8, 19]. They can support casual interactions and informal awareness in work environments because it is easy to notice a person's whereabouts when they have a TR embodiment [5]. In home settings, they can allow distance-separated family members to share activities together [8]. Sometimes the ability to 'bump into' objects has been shown to be beneficial and a means to enhance one's feeling of remote presence (such as in a home environment) [8]; however, in many situations it can be highly problematic and socially awkward to run into objects or people using a TR, such as in an academic conference or work setting [1].

Despite the benefits, TRs are subject to operational challenges, as robots are often shared asynchronously amongst users in work and conference settings; thus, no one individual owns the "embodiment" (the look and sound, e.g., [5, 20, 21]) and remote users are unable to customize the robot's appearance unless physical items are attached by a user who is local to the robot [22]. Privacy issues sometimes arise for TR users because they must appropriate themselves for two different environments simultaneously: their own local environment as well as the remote one [1, 22]. The spatial relationship between the robot and the environment can also be difficult to understand [1, 22]. This can be especially difficult when remotely operating TRs in unknown spaces [1]. Next to the unknown spatial layout and one's own position and (spatial) orientation (as represented through the TR), the dynamic nature of an environment can make navigating TRs even more challenging. For example, both the movement of objects and their density in the environment, e.g., spaces crowded with people, can make navigation difficult [2]. While researchers have focused on increasing the user's spatial orientation and awareness in the remote space, e.g., through wide angle cameras [10, 11] or the usage of sound feedback [11], navigation often is still limited as the amount and kinds of feedback the users receive remain constrained when compared to how we navigate as human's in the real world. So, generally, the amount of cues one receives may not be enough to maintain an adequate level of spatial awareness, in order to interpret a situation and mentally project towards its future status [23] to navigate around effectively.

There is a variety of research that explores feedback for teleoperation situations. For example, researchers have looked into using haptic feedback to improve accuracy and awareness [24] and navigation via collision avoidance [25, 26]. Lee et al. [27] explored the performance of haptic feedback on navigation performance with a mobile (non-telepresence) robot and found benefits. Studies of self-motion have similarly explored feedback in virtual reality settings. For example, visual cues [28], sound cues [29], and foot steps [30] have been tested as forms of feedback and shown value. Our approach, in contrast, does not rely on feedback per se, but instead modifies the speed of the TR directly to avoid the need for such feedback.

Several methods for automatic SC have been developed in different fields, including Virtual Reality and the automotive sector. To automatically control the speed of a viewer in a virtual environment, Mackinlay et al. scaled the distance from the viewer to a target to determine the movement speed [12]. Ware and Fleet developed this idea further and presented a method that considers the distances to all visible points [13]. They found that the minimum distance works best to determine the ideal speed, but that the average was also competitive. The usage of sensor-data to control velocity is widely used in commercial vehicles, and generally known under the name adaptive cruise control. These systems often rely on some form of adaptive control system [14]. Within the frame of these systems, research among others has focused on autonomous throttle and brake actuation [31, 32], break system modeling [33], stop-and-go mechanisms [34] and distance control [15]. The control of autonomous vehicles shows great

resemblance with these systems, and is usually comprised of perception, decision, and control components to drive a vehicle. The speed controller of an autonomous vehicle thereby is often based on two control levels: the higher level that deals with acceleration, and the lower level that controls the throttle and brakes. Models that drive the SC can be self-adaptive over time, based on learning methods [35, 36]. DJI Drones (https://www.dji.com) include an automatic deceleration and stop feature when there is a obstacle in front of them. Similarly, humanoid robots, such as the Pepper robot, have proximity sensors which allows such robots to slow down and adjust the speed of the device based on human interaction through a Gaussian Mixture Model [37]. The purpose of these robots are different from TRs; TRs are designed to facilitate interaction with people, with an aim to provide a remote presence with visual and auditory feedback to the operator and the people they interact with. Hence, results from autonomous robot control are not directly applicable to our experiment here, where the user interactively controls the TR and interacts with other people.

Most recently, shared control algorithms were used as assistance systems to control the speed of TRs [17, 18]. Yet, the proposed methods increased the task execution time and were not tested in dense and social environments. The SC system we implemented resembles to a degree the SC methods used in autonomous vehicles, as we semi-autonomously adjust speed based on the context around the TR. Yet, in contrast to autonomous SC methods, our method has to take into account that the user interactively navigates the TR.

We used a Beam+ TR (https://www.suitabletech.com/products/beam) in this study, as in Fig 1. We also used an attached Arduino Mega, a Raspberry Pi 3, and six out of twelve

(a)          (b)

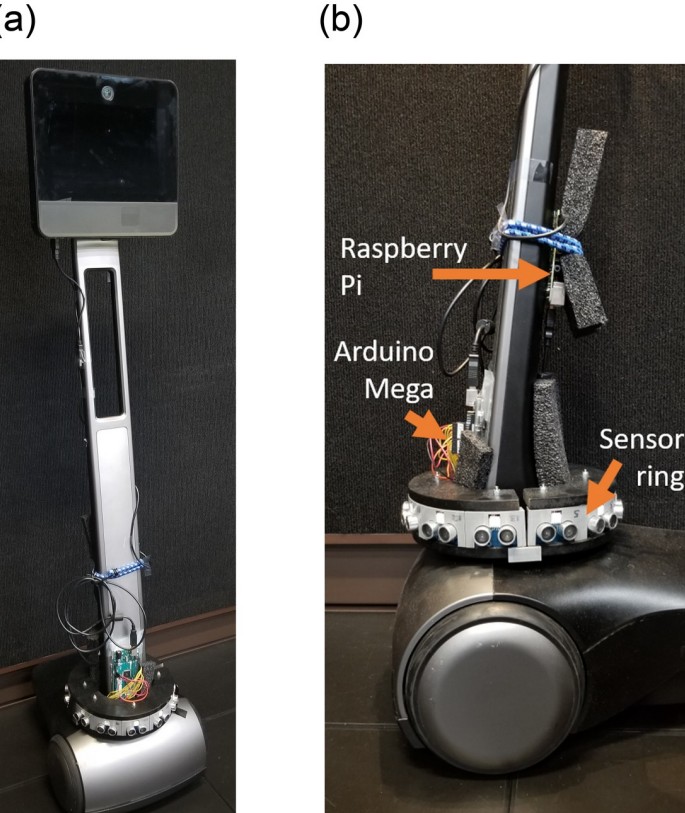

**Fig 1. (a) Beam TR used in the study, (b) additional hardware components mounted on the base of the Beam.**

ultrasonic distance sensors on the device, as illustrated in Fig 1. This apparatus had been designed and previously used in different work [38]. there is no overlap in terms of research topic, software, hypotheses, analyses, results, findings and contributions between that work and our current one, as the authors of the previous work used the distance sensor ring to measure the distance around the TR to give haptic feedback to the users' feet [38]. While this setup was designed for a different research purpose, we developed our system based on the previously developed hardware. For our work, we altered the software running on the Arduino and Pi to support a specific distance sampling schema that suits the needs for SC algorithms. In this document, we still describe the hardware in section 4, as it is directly relevant to our work. Yet, we explicitly state that the sensor ring was not designed and built by us nor is a contribution of this work.

## 2 Motivation

As noted before, driving a TR can be challenging, which can be caused by a variety of factors. For example, the cameras often restrict what can be seen around the TR, and often causes distance estimation problems. As a result, especially in dense and potentially highly dynamic scenes, it will be difficult to gather and maintain situational awareness around the TR [39]. To alleviate driving challenges and inspired by autonomous vehicle control paradigms that rely on multi-directional sensing capacities that also sense in directions not directly covered in the camera view, we chose to explore the use of TR driving aids. Yet, instead of fully autonomous control, we want to provide users with a suitable level of control to not reduce their sense of control and agency, but without needing to provide additional feedback through the UI.

### 2.1 Hypotheses

Our hypotheses are directly connected to previous work, which showed that automatically adapting a TR's speed lowers the operator's cognitive load [16, 17], helps people to avoid obstacles [40], and decreases the number of collisions [16]. These studies neither tested SC algorithms in static dense environments nor focused on their effect during social interaction. To assess the usefulness of the implemented distance-based SC methods, we formulated the following hypotheses, which we investigated in two user studies.

**H1**. **Distance-based speed control improves TR navigation behaviour**: We predicted that automatically slowing down the TR as it moves closer to potential obstacles and other objects would allow users to navigate more safely and avoid collisions more effectively. Similar to other domains [12] and to autonomous assistance research with TRs [16–18, 40], we expect to see better performance with a SC algorithm.

**H2**. **Distance-based speed control also increases the user's spatial presence in the remote environment**: We hypothesized that automatic SC would indirectly improve users' sense of spatial presence and situational awareness of their remote surroundings, by allowing them to focus less on the challenge of navigation, thus freeing up mental resources to be more present and aware of their remote surroundings.

While assessing H1 and H2, we also explore user reactions to the automatic SC algorithm, through interviews and questionnaires after the study tasks. Previous work on TRs had shown that assistance through SC improves TR navigation behaviour in term of number of collisions in static environments (at the cost of increased task times) [17, 18] and decreases cognitive load [17]. Yet, interestingly, SC algorithms have not been evaluated in an environment where a user has to remotely interact socially, which is a very common use case for TRs.

To test the effect of the distance-based SC method on TR navigation behaviour, we evaluated it first in a tightly-controlled static environment with both narrow passages as well as wide corridors (Study 1). To increase the ecological validity of our results, we then investigated in Study 2 the user experience in a conference-like setting with social interaction tasks.

## 3 Telepresence robot, apparatus and software used in this study

### 3.1 Distance sensors and data acquisition

In this study, we used hardware that had been designed for another research project [38]. This approach allows us to demonstrate that SC can be applied to different TRs, even if they cannot be modified or have been designed with SC in mind. Their work mounted a ring of twelve ultrasonic distance sensors onto the "neck" of the Beam+ TR. These equally-spaced 40 kHz ultrasound sensors were connected to the analog pins of an Arduino Mega, and are able to reliably detect objects at distances between 2 and 120 cm in front of them. With this setup, obstacles around the TR could be located within a 30˚ cone for each sensor. Here, to increase the data acquisition rate we did not use all of the available sensors. Instead, and since we were only interested in forward motions, we collected data from the sensors at the front of the ring, which corresponds to a 180˚ field of "view". To further increase the sampling frequency and to reduce the interference between sensors, we collected data only from alternating sensors in the sequence within any given sampling interval. In other words, we collected data in the following order: the first, fourth, second, fifth, third, and sixth sensor, and we repeated this sequence. As a result, we achieved a 40 Hz (25 ms) data process rate. This rate proved to be sufficient for the implemented SC method within the environments used in our user studies.

Since the Arduino Mega provides a sufficient number of input pins and provides libraries to convert analog ultrasound distance readings to digital information, we use it as a 'bridge' between the sensors and the Raspberry Pi. The software running on the Arduino software thus only manages the analog sensor data acquisition and converts this information to digital data that is sent to the Raspberry Pi through a serial link.

The data received by the serial link is forwarded to the desktop computer by the Raspberry Pi through the OSC (Open Sound Control) library and User Datagram Protocol (UDP) communication with Python 3.4 code. The Raspberry Pi was connected to the university's wireless network.

### 3.2 Desktop computer

We used a PC with an Intel (R) Core (TM) i7-5890 CPU with 16 GB RAM and a NVIDIA GeForce RTX2080 graphics card. A BenQ 27" HD HP desktop monitor and, to control the Beam+, a Logitech keyboard as well as a Xbox One gamepad were connected to the computer.

### 3.3 User interface and user interaction

For the GUI, participants interacted through the regular TR interface provided by the Beam. We did not change or alter the features in this GUI, as such modifications are not supported by the Beam manufacturer. A sample screenshot of the GUI is shown in Fig 2.

The GUI of the Beam is designed to show two camera views: a forward-facing camera view, which is mostly used to socially interact, and a downward facing camera, which is mostly used to navigate in the environment. This GUI only allows users to change the split of the (single) window between the downward-facing and forward-facing camera video streams. There are no default dimensions: users can adjust the camera view size based on their preference and the Beam software always stores and re-uses the latest video size setting. For the first study, we

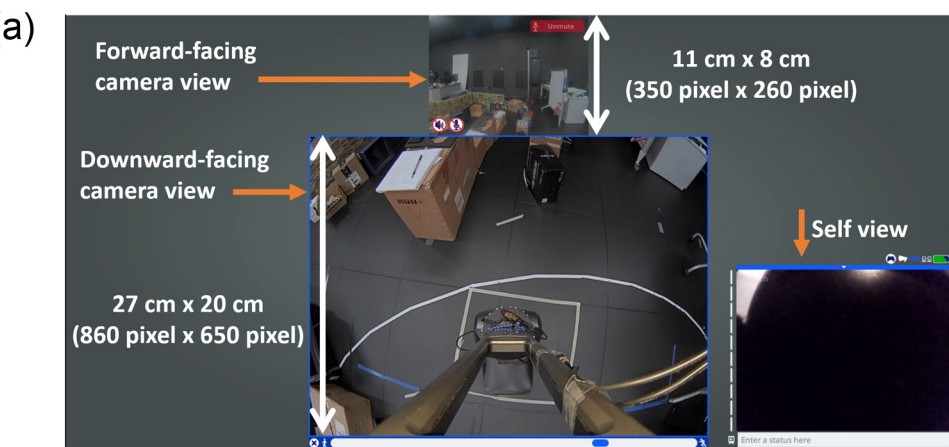

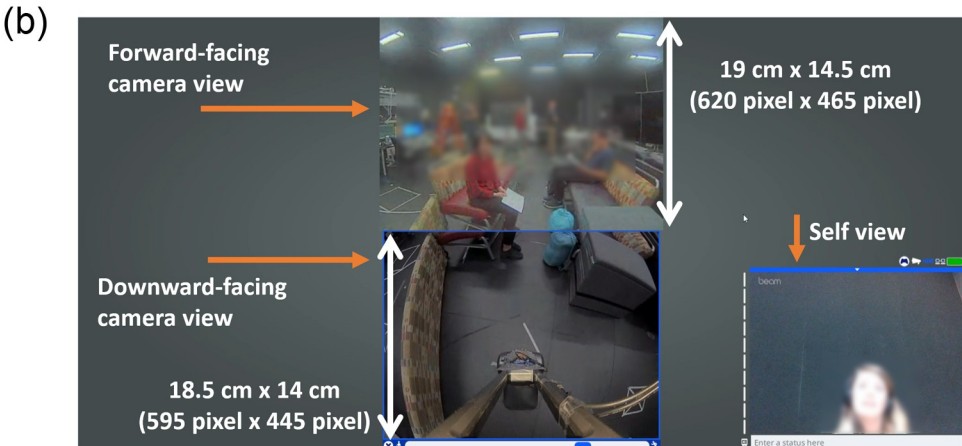

**Fig 2. Beam GUI Layouts during the experiment.** Layout for (a) User Study 1 and (b) User Study 2. Parts of images blurred for anonymity.

wanted users to only focus on the obstacles, so we enlarged the lower camera view to the maximum values allowed by the Beam GUI, which are 860 pixel x 650 pixel or 27 cm x 20 cm, as seen in Fig 2(a). In the second study, we wanted users to use both, i.e., the downward camera to navigate the obstacle course and the forward facing one to interact with other people. To allow users to focus simultaneously on social interactions and navigation, we allocated 620 pixel x 465 pixels or 19 cm x 14.5 cm to the forward-facing camera, as in Fig 2(b). In study 1, and since there were no actors interacting with the participant, the participant's self-view captured through the webcam of the computer was blocked with a piece of paper. In study 2, participants saw their self-view within the GUI to help them communicate with the actors.

We note that the optical flow in the camera due to the movement of a TR is already a form of feedback for the user: drivers can easily perceive how fast or slow the robot is moving by looking at the camera view(s). Thus, we did not provide any additional feedback mechanism for the speed beyond the existing camera views of the remote environment.

For the driving interaction, we used either a regular keyboard or a Xbox One gamepad to control speed and direction. While participants were using the keyboard, they pressed arrow keys with their dominant hand. In this condition, the SC is Boolean (ON/OFF), i.e., the robot is either moving a constant speed forward or stops. More specifically, the UP/DOWN arrow key

corresponds to forward/backward movement at a constant speed. When participants were using a gamepad, they operated the left joystick with their thumb to control the TR device. In this condition, participants were able to alter the speed of the TR continuously with the joystick. In either condition, participants did not have to use other keys or buttons on the keyboard, respectively gamepad. To be able to compare the TR navigation behaviour for the two input devices objectively, we did not add any additional feedback to input devices, such as active feedback through vibrating the gamepad with the motion of the joystick.

### 3.4 Distance-based speed control

We implemented a SC algorithm with Python 3.4. The purpose of this software was to receive the commands given by the user (originally destined to be sent directly) to the Beam software and to alter them according to the distance between the TR and the obstacle(s). For this we used pynput and the Python keyboard library to intercept speed-related keyboard input events and altered these events as specified below, so that the Beam software receives input that corresponds to the speed specified by the result of the distance-based SC method. For the gamepad, we used the pyvjoy library which maps all joystick input to the [0-1] range, with 0.5 corresponding to the neutral position.

The software was running in an infinite loop. In each iteration, we updated the incoming distance data from sensors sent by Raspberry Pi, ran the distance-based speed algorithm, and sent commands to the TR to to modify its speed during steering. It took about 23 ms (43 Hz) to run each loop. Thus, on average, the data for each of the 6 sensors was updated every 138 ms ($\approx$ 7 Hz). This was not pre-chosen, but was the highest average update rate we could reach with all hardware, software, and networking-induced delays. Previous work on remote surgery studies had identified that the average delay should be less than 700 ms in their application [41]. Our application scenario does not have the same life-or-death criticality as surgery, nor does it have the same millimeter-accuracy requirements. Since we cannot increase the data acquisition rate, and this data rate is larger than required for precise tasks, we believe that our update rate is acceptable. Moreover, we asked our participants after the studies if they had observed any significant delays that might have affected their driving performance, but no one made negative comments about the potential effects of delays.

While there are various possibilities for the SC methods, algorithms, and user interfaces [16–18], we implemented a distance-based SC algorithm that works as a middle-ware between the user and TR GUI. Since Beam+ is not an open system, we did not, and in fact could not, change or alter the code or hardware of the Beam+. We also did not include additional GUI elements, including visual and auditory feedback to the user, since not all TR GUIs use the same interface. Moreover, we wanted this setup to be applicable to other TR systems to improve generalizability of the results.

For the SC, we experimented with different distance/speed curves, such as a linear, exponential, or logarithmic model. We also tried a PID (proportional, integral, derivative) controller, which is commonly used in the control systems work [42, 43]. In our pilot trials, we observed that a PID controller did not work well, since the time interval between two distance samples at full speed was too large. To improve data acquisition, we used only half the available sensors, and, through various code optimizations, such as using multi-threading maximized the sampling rate within the given hardware platform. Yet, even this was not good enough to robustly drive the TR with a PID controller or derivative algorithm, and we thus did not use this approach. After optimizing the system as far as possible, the final update rate for our system was $\approx$ 7Hz for the distance sensors, which included all hardware, software, and networking delays. Additionally, when a command was sent to the motors, there were delays due to

other factors, such as network ping. Yet, as mentioned above, we were limited to control our system through manipulating the input stream of the commercial (closed) product, which limited our technical options and the control frequency.

Before starting the experiments, we tested the SC algorithm in pilot studies with 2 experts with more than 2 years of TR driving experience and 5 non-expert users. None of these individuals participated in the main experiments. To create a movable obstacle, we used a chair with 4 legs on casters and put duct-tape at the height of the sensors. In this pilot setup, if the SC algorithm failed, the TR hit the chair, and since the chair was movable, the TR moved the chair around. This allowed pilot participants to try different SC algorithms without causing damage. We did not observe a notable performance difference between linear, exponential, and logarithmic models in this pilot. Pilot participants also did not seem to notice differences between the algorithms. Given that we did not observe big differences and pilot participants did not prefer a specific SC algorithm, we chose a linear SC mapping for simplicity, i.e., *SystemInput = UserInput − distance * 0.0025*, as it was the most robust option and worked well with our limited data acquisition rate.

We placed the sensors around the bottom of the "neck" of the TR, at 30 cm above the ground, as shown in Fig 1. To deal with situations where the ultrasound sensors did not yield sufficiently accurate information, e.g., when confronted with gaps between chair legs or metal surfaces, we used duct tape to cover the corresponding space at the level of the sensors.

The *UserInput* corresponds to the input speed value given by the user to the system. For both gamepad and keyboard, the maximum input value was set to be 0.85 m/s through the GUI of the Beam, which corresponds to a leisurely walking speed, similar to how one would walk when engaged in a conversation. We checked this speed with the distance sensors on the TR and approved it. We also tested this speed in other pilot studies, where participants were walking next to the TR and subjects found this level of speed adequate for the context. This method has been used in other walking speed and TR research, e.g., [44, 45].

As mentioned in the Apparatus section, there are six sensors facing forward on the sensor ring, each of which covers a 30° cone. To compute the distance, we average the distances from the front two sensors of the Beam+ as the forward distance. We then use this value as *SystemInput* to control the speed of the TR (through the Beam software). To achieve stable results, we only activated the SC algorithm at distances between 10 and 120 cm. The suggested highest distance to correctly measure distance with the used ultrasonic sensors is 120 cm. Since the experiment aims to study TR navigation behaviour in narrow and dense spaces, we did not modulate the speed in sections where all obstacles are far away and 120 cm was enough for this purpose. Also, if the distance between the sensor and any obstacle was less than 10 cm, the TR is either very close to hitting an obstacle or has already hit it. Since we wanted the TR not to stop but only to move slowly at distances less than than 10 cm, the control method shown in Fig 3 assisted users in moving the TR slowly when they were very close to obstacles, by thresholding the movement speed to 0.17 m/s, which is the minimum value for the motors of the used TR to move smoothly. At lower speeds, the TR jerks or does not respond to commands correctly.

The resulting distance/speed curve of the overall system is shown as Fig 3. If the TR gets close to objects, but the objects are to the side, we also reduce the speed of the robot to 0.17 m/s based on the minimum distance received from the four side sensors. This allows the user to maneuver in a dense environment in situations where there is no obstacle in front of the TR. We also included a method to decrease jerky movements. When the TR is commanded to change the speed, we did not immediately increase or decrease the speed of the device, but did it in two steps to eliminate jerky movements. For instance, while the TR was stationary, if the user gave the full speed command, we first used an intermediate speed of 0.51 m/s and only

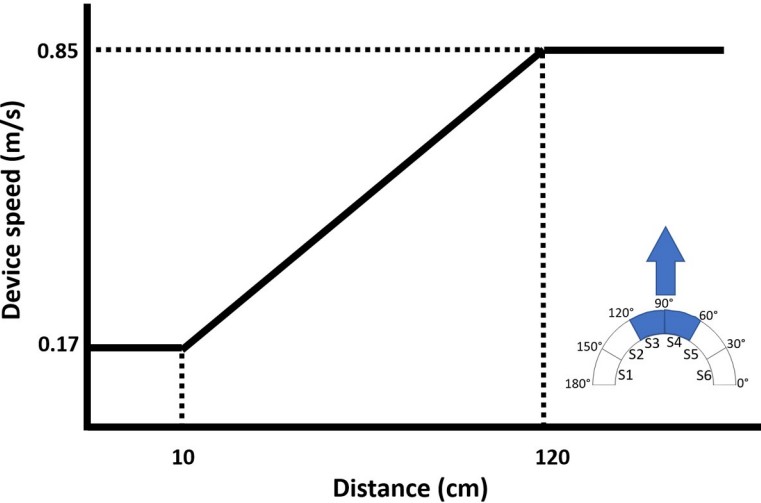

**Fig 3. Distance-based speed control mapping for the speed of the TR.**

sent the final speed of 0.85 m/s after $\approx$ 138 ms. Similarly, when the device got suddenly closer to objects, e.g., when rounding a corner or edge with obstacles immediately after, we did not decrease the speed immediately, but averaged it with the previous input. This helped us to reduce unexpected and unnatural sudden speed changes which can distort the equilibrium of the device, i.e., can lead to jerky movements and forward/backward wobbling of the TR. With the graded transitions, this issue did not occur.

## 4 User study 1

In this first study, we designed a static dense environment with milestones that represent various challenges that could be found in dense environments and investigated how user navigation behaviours and their experience are affected by the implemented SC method in an obstacle course with different input devices. In our context, increased density corresponds to a higher amount of objects with smaller distances between them in the environment.

### 4.1 Experimental setup

To investigate the effectiveness of the SC algorithms in typical conference-like dense situations where users have to navigate in narrow and wider areas, we created a dense environment as shown in Figs 4 and 5 and asked users to drive through it while avoiding collisions.

Figs 4 and 5 illustrate the same environment. The dark pink box (bottom right corner in the photo) indicates the start and finish area for the experiment. We first divided the obstacle course into 42 smaller milestones **M1-M42**, as shown in Fig 5. Between every milestone, TR had to accomplish a specific small-task, such as turning 90˚ in a tunnel (e.g., **M3-M4**, **M12-M13**, **M34-M34** and **M23-M24**), turning 90˚ in open-space (e.g., **M6-M7**, **M8-M9**, **M10-M11** and **M36-M37**), going underneath the ladder (**M19-M20** and **M27-M28**) or going straight in a tunnel (**M11-M12** and **M35-M36**). These milestones were used to collect data in detail. To ease the data analysis, and based on the number of collision observations, we also divided the movement of the TR into 8 different segments, labelled **S1-S8** in both Figs 4 and 5. When we were designing the obstacle course, we tried to include sections that correspond to various real life cases. We used empty boxes and other non-critical objects to create an obstacle course that was safe to drive through for both the TR and the environment. Since the motors

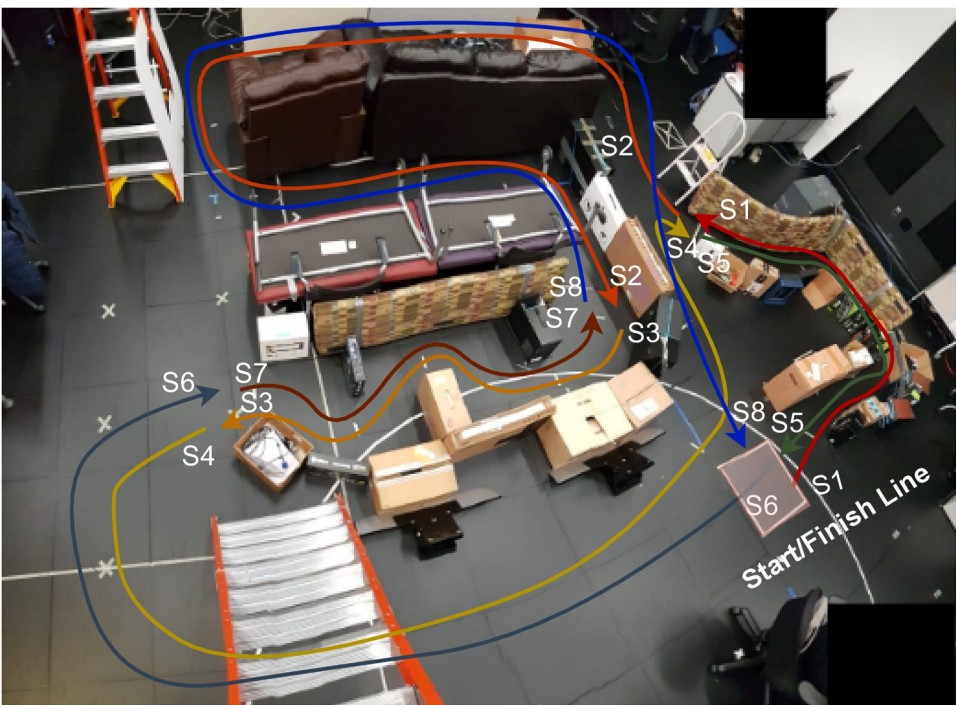

**Fig 4. Real world view of the obstacle course.** This photo was taken from top of the ladder as a panoramic picture. Distortions in the picture caused by the panoramic stitching.

of the Beam+ TR are fairly powerful, it was possible to dislocate even heavy objects with it. Thus, we designed our obstacle course to make it a safe environment and to avoid the potential for damaging the TR or other equipment. We picked objects, such as carton boxes, that made it easy to reset any portions of the obstacle course, if participants collided with and thus moved any of the objects that comprised the obstacle course. We also taped such objects to fill gaps between obstacles at the level of the sensors in the constrained path segments. Since the ultrasound sensors can return different distance values for different materials, we also used the tape as a uniform "reflector" material to address this issue for those objects that had notably different reflective properties. During the experiment, the experimenter fixed any re-located portions of the previous part of the obstacle course when the TR reached the next milestone.

## 4.2 Participants

Twelve participants (10 female, average age of 21.9, SD 1.9) participated in our experiment. All participants were right-handed and had never used a TR before.

## 4.3 Procedure

This work involved a user study (Human Subject Research), conducted with approval of the Simon Fraser University Research Ethics Board (REB [2015s0283]). All participants signed informed consent forms and their data were analyzed anonymously.

Before starting the experiment, the experimenter explained the task and asked participants to follow the experimenter in a physical walk-through of the obstacle course, using the same path as for the TR. We did this to familiarize them with the navigation task and spatial layout of the environment, and ensure that they knew the correct path before starting the actual

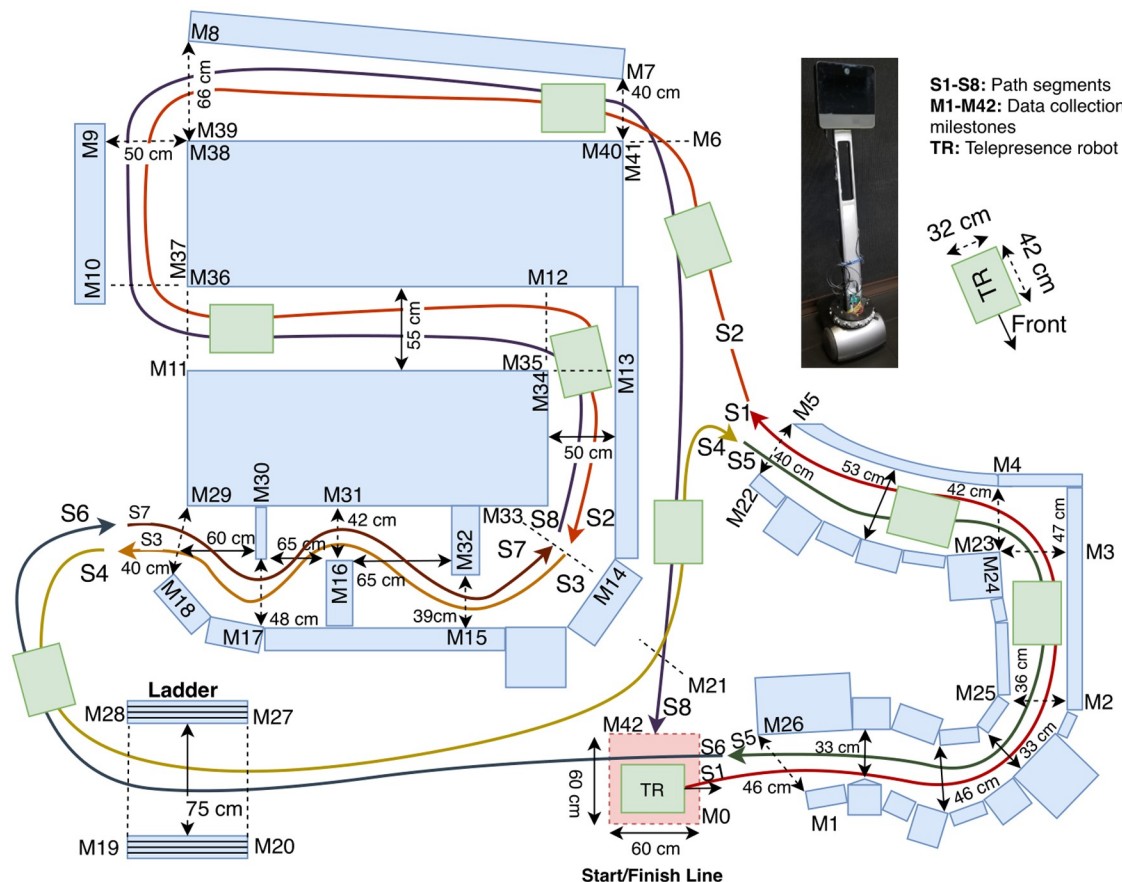

**Fig 5. The obstacle course designed to evaluate TR maneuvering in dense environments.** For clarity, only the distances for the most important parts of the path are shown in the figure. Dashed lines show the milestone lines. The colored lines correspond to the whole sequence of navigation (red → orange → yellow → green → dark blue → brown → light blue).

experiment. After pilot studies we decided on this procedure as our main objective was to investigate participants' TR maneuvering behaviours and not their way-finding skills.

After walking the path, we asked participants to complete a demographic questionnaire, to collect data about their age, gender, gaming experience, and if they have a driver's license. Then, we asked them to experiment with the TR in an open space, so they would get used to the interface, until they got comfortable with driving the TR. To make the driving task more realistic and to enable participants to focus on driving in the experiment, participants were given a map of the obstacle course, so they did not have to "search" for the path. They also completed a single iteration of the task (see below) through the obstacle course, where they experienced the four different conditions (keyboard or gamepad, with and without the SC for each input device), in the four main sections of the course. This enabled them to learn the layout of the environment. When the participants felt ready to start the experiment, we asked them to move the TR to the start/finish area shown in Fig 5. Based on our observations, participants took an average 5 minutes to complete the training phase. We saw no notable differences in terms of learning between the different conditions, which is not surprising given that our participants had not driven TRs before.

Before participants started to drive the TR through the obstacle course in the evaluation session, we started the screen recording. We recorded the Beam GUI at 30 Hz to enable later

analysis of the movement of the TR. Participants had to follow the path segments from **S1** to **S8**. A trial ended when the TR crossed the boundaries of the finish line. To simulate a realistic telepresence setting, we blocked the participant's view of the obstacle course and the TR with large white large carton sheets so they never saw the TR or obstacles course directly during the main study.

After the experiment, participants filled a questionnaire, where we asked participants about their preferred driving method (with or without SC) and input method (gamepad or keyboard). We also asked them open-ended questions, such as, the reasons behind their preference of driving method and other comments. Further, we used 7-point Likert questions to investigate the ease of interaction of each condition, i.e., with and without the SC driving method, with the keyboard or the gamepad. We also asked if they thought that SC improved their TR navigation behaviour in terms of task completion time, hitting objects, and finally their physical and mental fatigue after the experiment.

During each task, an experimenter was in the obstacle course, and fixed any re-located or hit objects/boxes after the TR passed them. To enable this, we marked all object positions with black tape on the ground. Since the TR had to drive through each path segment in both directions, there was a need to fix the boxes as soon as possible (before the TR traversed the same segment again in the other direction). The experimenter also assisted participants through voice feedback if they got confused or deviated from the designated path. The logged data for any such episodes was manually removed before the main analysis process.

## 4.4 Experimental design

Each of the 12 participants performed 4 trials total, consisting of a factorial combination of two **Input Devices** ($ID_2$: Keyboard and Gamepad) × two **Speed Control** conditions ($SC_2$: ON and OFF). The order of trials was counterbalanced across conditions using a two-dimensional Latin Square design to avoid potential learning effects. In total, the experiment took about 40 minutes for each participant.

By using the video recorded off the Beam GUI, we were able to collect timing data (in seconds) for each instance when the TR passed each milestone in Fig 5. We also counted the number of hits that occurred on both the front and the back of the TR as **Collision Side** ($CS_2$: Front or Back). We further divided these hits into two different **collision categories**: while looking at the the video, if the TR physically "bumped" into an object but did not dislocate it, we classified this as a "touch". If the TR bumped an object hard enough to dislocate it, we recorded this as a "hit" ($CC_2$: Hit and Touch).

## 4.5 Data analysis

The data were analyzed using $2_{ID} \times 2_{SC}$ repeated measures (RM) ANOVAs for the independent variables Input Device and Speed Control, with $\alpha = 0.05$ in SPSS 24. We used the Sidak method for post-hoc analyses. For non-normal distributions we used the ART method [46]. All detailed results, such as tables and figures for each dependent variable, can be found in the appendix. Fisher's test results for study 1 are shown in Table 1. Means (M), Standard Deviations (SD), Standard Error of Means (SEM), 95% Confidence Intervals (CI) vales are shown in Table 2. We also included the results for task completion time in Fig 6, average number of collision in Fig 7, average number of front touch in Fig 8, average number of back touch in Fig 9, average number of front hit results in Fig 10, and average number of back hit in Fig 11.

**4.5.1 Task completion time.**   Completion time was normal after a logarithmic transformation (Shapiro-Wilk test result was $W(48) = 0.982$, n.s., Skewness = 0.249, Kurtosis = -0.422). The RM ANOVA results showed no significant main effects of SC or input device

**Table 1. Fisher's test results for study 1.**

| Dependent Variable | Speed Control | Input Device | Input Device x Speed Control |
|---|---|---|---|
| Task Completion Time | $F(1,11) = 4.126$, $p = 0.067$ $\eta^2 = 0.27$ | $F(1,11) = 0.647$, $p = 0.438$ $\eta^2 = 0.056$ | $F(11,1) = 0.201$, $p = 0.66$ $\eta^2 = 0.018$ |
| Total Number of Collisions | $F(1,11) = 36.75$, $p < 0.001$ $\eta^2 = 0.77$ | $F(1,11) = 1.986$, $p = 0.186$ $\eta^2 = 0.153$ | $F(1,11) = 0.011$, $p = 0.92$ $\eta^2 = 0.001$ |
| Front Touch | $F(1,11) = 21.78$, $p < 0.001$ $\eta^2 = 0.66$ | $F(1,11) = 0.152$, $p = 0.698$ $\eta^2 = 0.014$ | $F(1,11) = 0.31$, $p = 0.594$ $\eta^2 = 0.27$ |
| Back Touch | $F(1,11) = 3.160$, $p = 0.103$, $\eta^2 = 0.233$ | $F(1,11) = 2.708$, $p = 0.128$, $\eta^2 = 0.198$ | $F(1,11) = 0.062$, $p = 0.807$, $\eta^2 = 0.006$ |
| Front Hit | $F(1,11) = 23.32$, $p < 0.001$ $\eta^2 = 0.014$ | $F(1,11) = 0.244$, $p = 0.631$, $\eta^2 = 0.022$ | $F(1,11) = 1.15$, $p = 0.25$, $\eta^2 = 0.118$ |
| Back Hit | $F(1,11) = 19.062$, $p < 0.001$ $\eta^2 = 0.634$ | $F(1,11) = 2.518$, $p = 0.141$, $\eta^2 = 0.186$ | $F(1,11) = 0.012$, $p = 0.913$, $\eta^2 = 0.001$ |

conditions, nor any significant interactions. Detailed results are shown in Fig 6, Tables 1 and 2. Results suggest that subjects might be faster with a gamepad when SC was turned off, but we could not identify any significant differences.

**4.5.2 Collisions.** Collision dependent variable was normal after a logarithmic transformation (Shapiro-Wilk test result was $W(48) = 0.962$, n.s., Skewness = -0.22, Kurtosis = -0.85). As illustrated in Fig 7(c) and 7(c), adding SC reduced the average number of collisions from $M = 6.36$, 95%-CI [5.52, 7.20] to $M = 3.69$, 95%-CI [3.01, 4.37] ($F(1, 11) = 36.75$, $p < 0.001$, $\eta^2 = 0.77$). We were not able to identify any significant main effect of input device, or any

**Table 2. Mean (M), Standard Deviation (SD), Standard Error of Mean(SEM) and 95% Confidence Intervals (CI) results for study 1.**

| Dependent Variable | General Results | Speed Control | | Input Device | | Input Device | | | |
|---|---|---|---|---|---|---|---|---|---|
| | | | | | | Gamepad | | Keyboard | |
| | | ON | OFF | Gamepad | Keyboard | Speed Control | | | |
| | | | | | | ON | OFF | ON | OFF |
| Task Completion Time (seconds) | M = 350, SD = 83, SEM = 12, 95%CI [326, 374] | M = 357, SD = 88, SEM = 18, 95%CI [320, 395] | M = 342, SD = 78, SEM = 16, 95%CI [309, 376] | M = 328, SD = 79, SEM = 16, 95%CI [295, 362] | M = 371, SD = 83, SEM = 17, 95%CI [336, 407] | M = 336, SD = 68, SEM = 20, 95%CI [293, 380] | M = 320, SD = 90, SEM = 26, 95%CI [263, 378] | M = 379, SD = 104, SEM = 30, 95%CI [313, 445] | M = 364, SD = 60, SEM = 17, 95%CI [326, 403] |
| Total Number of Collisions | M = 5.03, SD = 3.98, SEM = 0.28 95%CI [4.46, 5.59] | M = 3.69, SD = 3.35, SEM = 0.34, 95%CI [3.01, 4.37] | M = 6.36, SD = 4.13, SEM = 0.42 95%CI [5.52, 7.2] | M = 4.67, SD = 3.39, SEM = 0.34, 95%CI [3.98, 5.36] | M = 5.38, SD = 4.49, SEM = 0.45, 95%CI [4.47, 6.29] | M = 3.5, SD = 2.98, SEM = 0.43, 95%CI [2.63, 4.36] | M = 5.85, SD = 3.39, SEM = 0.49, 95%CI [4.86, 6.84] | M = 3.89, SD = 3.71, SEM = 0.53, 95%CI [2.81, 4.97] | M = 6.87, SD = 4.74 SEM = 0.68, \95CI [8.25 5.49] |
| Front Touch | M = 7.22, SD = 3.39, SEM = 0.49 95%CI [6.24, 8.21] | M = 5.8, SD = 2.74, SEM = 0.56, 95%CI [4.67, 6.99] | M = 8.63, SD = 3.46, SEM = 0.7, 95%CI [7.16, 10.08] | M = 6.1, SD = 3.02, SEM = 0.61, 95%CI [4.80, 7.35] | M = 8.4, SD = 3.42, SEM = 0.69, 95%CI [6.92, 9.82] | M = 4.91, SD = 2.83, SEM = 0.82, 95%CI [3.11, 6.72] | M = 7.25, SD = 2.83, SEM = 0.81, 95%CI [5.45, 9.04] | M = 6.75, SD = 2.41, SEM = 0.69, 95%CI [5.21, 8.28] | M = 10, SD = 3.59, SEM = 1.04, 95%CI [7.71, 12.28] |
| Back Touch | M = 7, SD = 3.71, SEM = 0.53 95%CI [5.92, 8.07] | M = 6.25, SD = 3.17, SEM = 0.64, 95%CI [4.9, 7.59] | M = 7.75, SD = 4.12, SEM = 0.84, 95%CI [6.01, 9.48] | M = 6.3, SD = 3.63, SEM = 0.74, 95%CI [4.79, 7.86] | M = 7.6, SD = 3.76, SEM = 0.76, 95%CI [6.07, 9.25] | M = 5.41, SD = 2.99, SEM = 0.86, 95%CI [3.51, 7.32] | M = 7.25, SD = 4.09, SEM = 1.18, 95%CI [4.64, 9.85] | M = 7.08, SD = 3.26, SEM = 0.94, 95%CI [5.01, 9.15] | M = 8.25, SD = 4.26, SEM = 1.23, 95%CI [5.53, 10.96] |
| Front Hit | M = 3.3, SD = 3.96, SEM = 0.57 95%CI [2.18, 4.48] | M = 1.04, SD = 1.48, SEM = 0.3, 95%CI [0.41, 1.67] | M = 5.62, SD = 4.36 SEM = 0.89, 95%CI [3.78, 7.46] | M = 3.2, SD = 3.06, SEM = 0.62, 95%CI [1.91, 4.5] | M = 3.45, SD = 4.7, SEM = 0.97, 95%CI [1.44, 5.47] | M = 1.41, SD = 1.78, SEM = 0.51, 95%CI [0.28, 2.55] | M = 5, SD = 3.07, SEM = 0.88, 95%CI [3.04, 6.95] | M = 0.66, SD = 1.07, SEM = 0.31, 95%CI [0.01, 1.34] | M = 6.25, SD = 5.43, SEM = 1.56, 95%CI [2.8, 9.7] |
| Back Hit | M = 2.56, SD = 2.3, SEM = 0.3 95%CI [1.88, 3.23] | M = 1.66, SD = 1.85, SEM = 0.37, 95%CI [0.88, 2.45] | M = 3.45, SD = 2.44, SEM = 0.49, 95%CI [2.42, 4.49] | M = 3.08, SD = 2.44, SEM = 0.49, 95%CI [2.04, 4.11] | M = 2.04, SD = 2.13, SEM = 0.43, 95%CI [1.13, 2.94] | M = 2.25, SD = 2.26, SEM = 0.65, 95%CI [0.81, 3.69] | M = 3.91, SD = 2.42, SEM = 0.70, 95%CI [2.37, 5.46] | M = 1.08, SD = 1.16, SEM = 0.34, 95%CI [0.34, 1.82] | M = 3, SD = 2.48, SEM = 0.71, 95%CI [1.42, 4.58] |

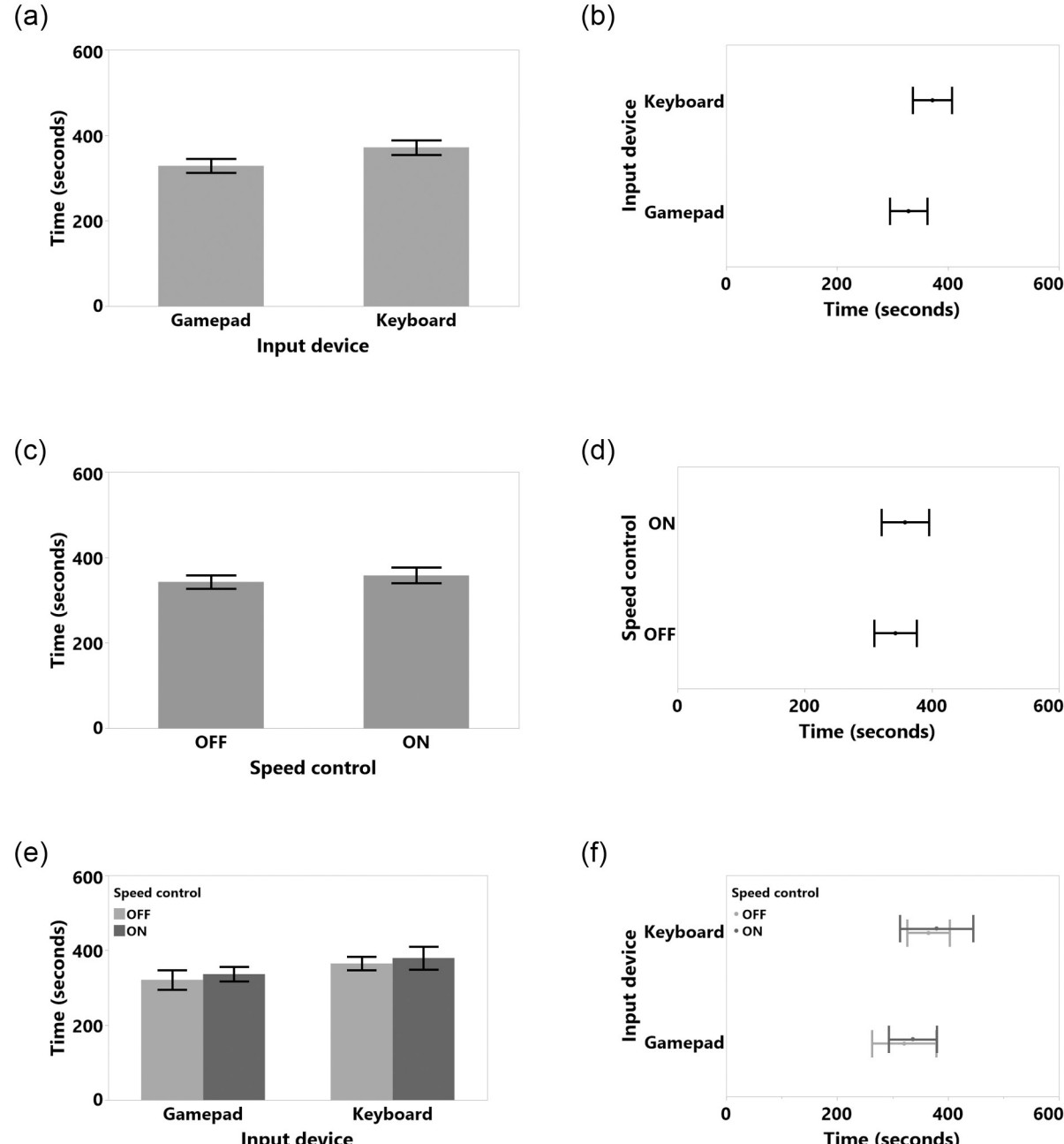

**Fig 6.** Task completion time means and standard error of means for (a) input device, (c) speed control, (e) input device and speed control interaction. Task completion time means and 95% confidence intervals for (b) input device, (d) speed control, (f) input device and speed control interaction.

interactions with SC. The results suggest that participants collided less with a gamepad, but the difference was not significant. Detailed results are shown in Fig 7, Tables 1 and 2.

**4.5.3 Detailed collision analysis.** Only the Front Touch collision dependent variable was normal after log transformation (Shapiro-Wilk test result was $W(48) = 0.953$, Skewness = 0.55, Kurtosis = -0.31). According to the results for front side touches (Fig 8(c) and 8(d)), SC significantly reduced the number of such collision ($F(1,11) = 21.782$, $p<0.001$, $\eta^2 = 0.664$) from

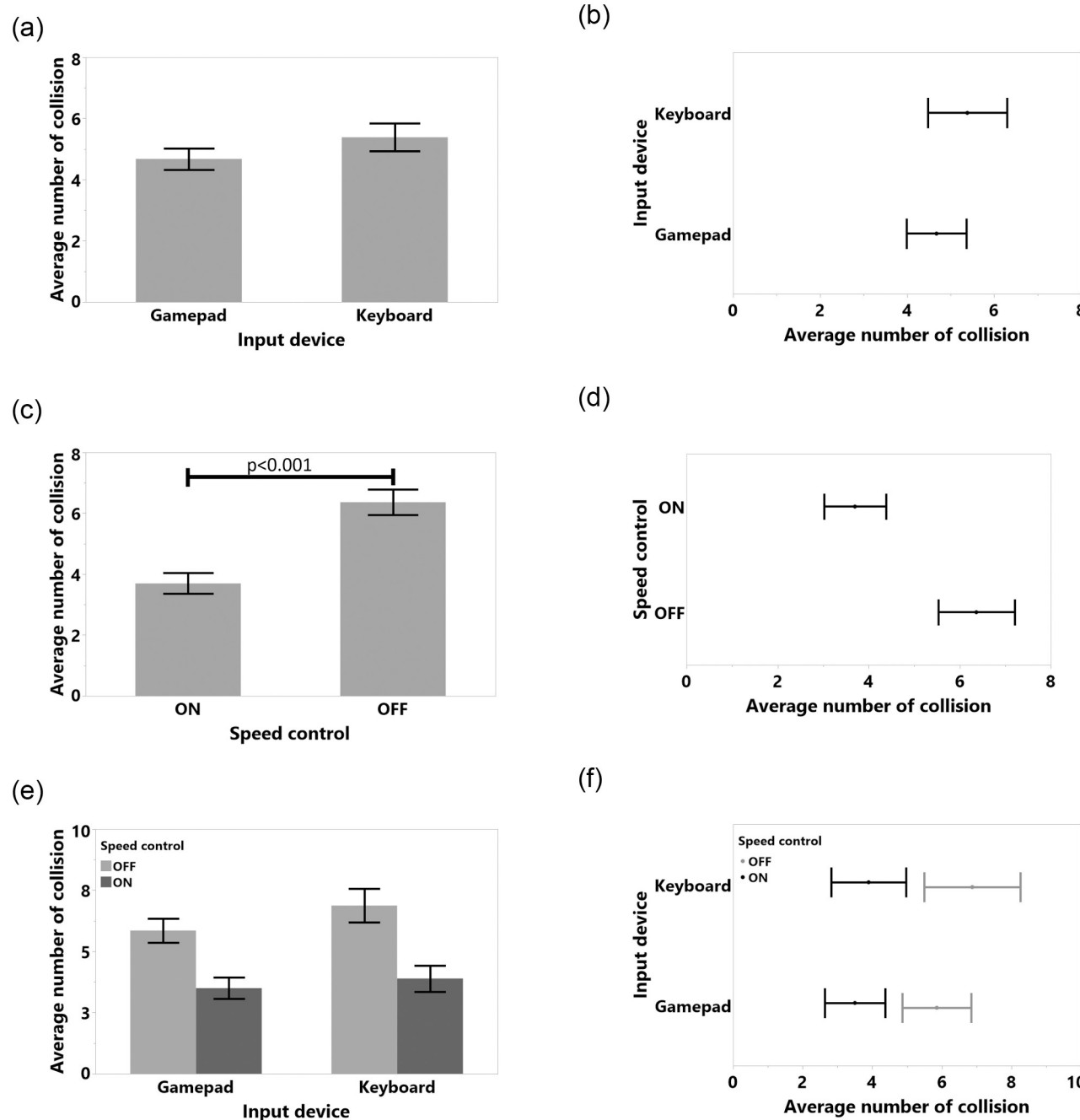

**Fig 7.** Means and standard error of means for average number of collisions for (a) input device, (c) speed control, (e) input device and speed control interaction. Means and 95% confidence intervals for number of collisions for (b) input device, (d) speed control, (f) input device and speed control interaction.

$M = 8.62$, 95%-CI [10.08, 7.16] to $M = 5.83$, 95%-CI [4.67, 6.99]. Moreover, SC significantly reduced the number of hits with the front side of the TR ($F(1,11) = 23.318$, $p<0.001$, $\eta^2 = 0.679$) from $M = 5.62$, 95%-CI [3.78, 7.46] to $M = 1.041$, 95%-CI [0.41, 1.67] as shown in Fig 10(c) and 10(d). Even though we did not acquire distance data at the back part of the TR and thus our system could not avoid hits that might occur at the back of the TR, SC significantly

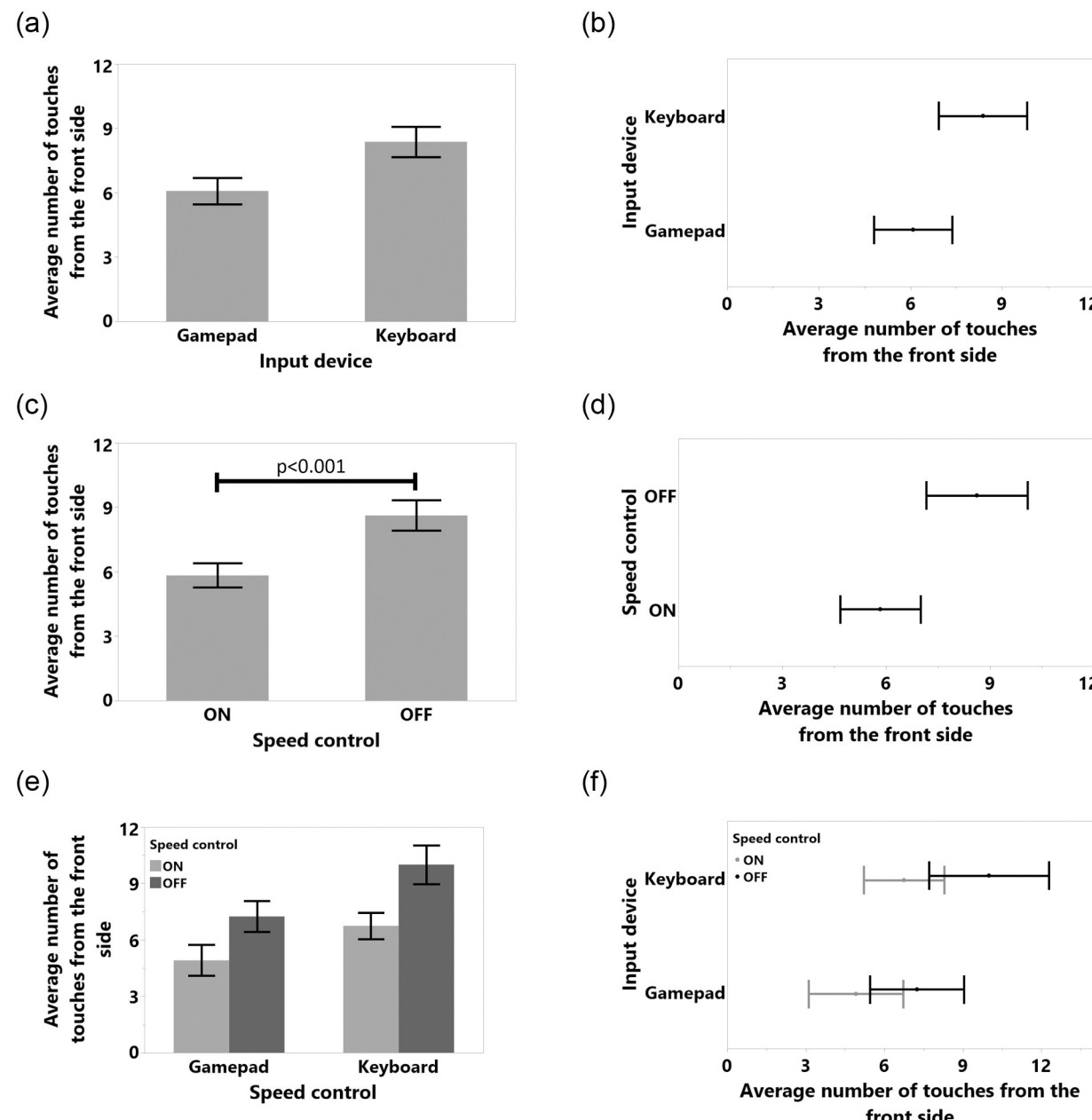

**Fig 8.** Means and standard error of means for number of touches from the front side for (a) input device, (c) speed control, (e) input device and speed control interaction. Means and 95% confidence intervals for number of touches from the front side for (b) input device, (d) speed control, (f) input device and speed control interaction.

reduced the number of hits with back part of the device from (F(1,11) = 19.062, p<0.001, $\eta^2$ = 0.634) M = 3.45, 95%-CI [2.42, 4.49] to M = 1.66, 95%-CI [0.88, 2.45] in Fig 11(c) and 11(d).

**4.5.4 Detailed analysis of segments and milestones.** In this study, we recorded 966 collision points. Only a total of 60 collisions by 12 participants occurred in **S2**, **S4**, **S6**, and **S8**. When we analyzed the data for these segments, we did not find any significant quantitative results to report. However, we observed that when the SC algorithm was activated in **S8**, between **M39**-**M40** where the tunnel gets narrower, participants adjusted their steering to

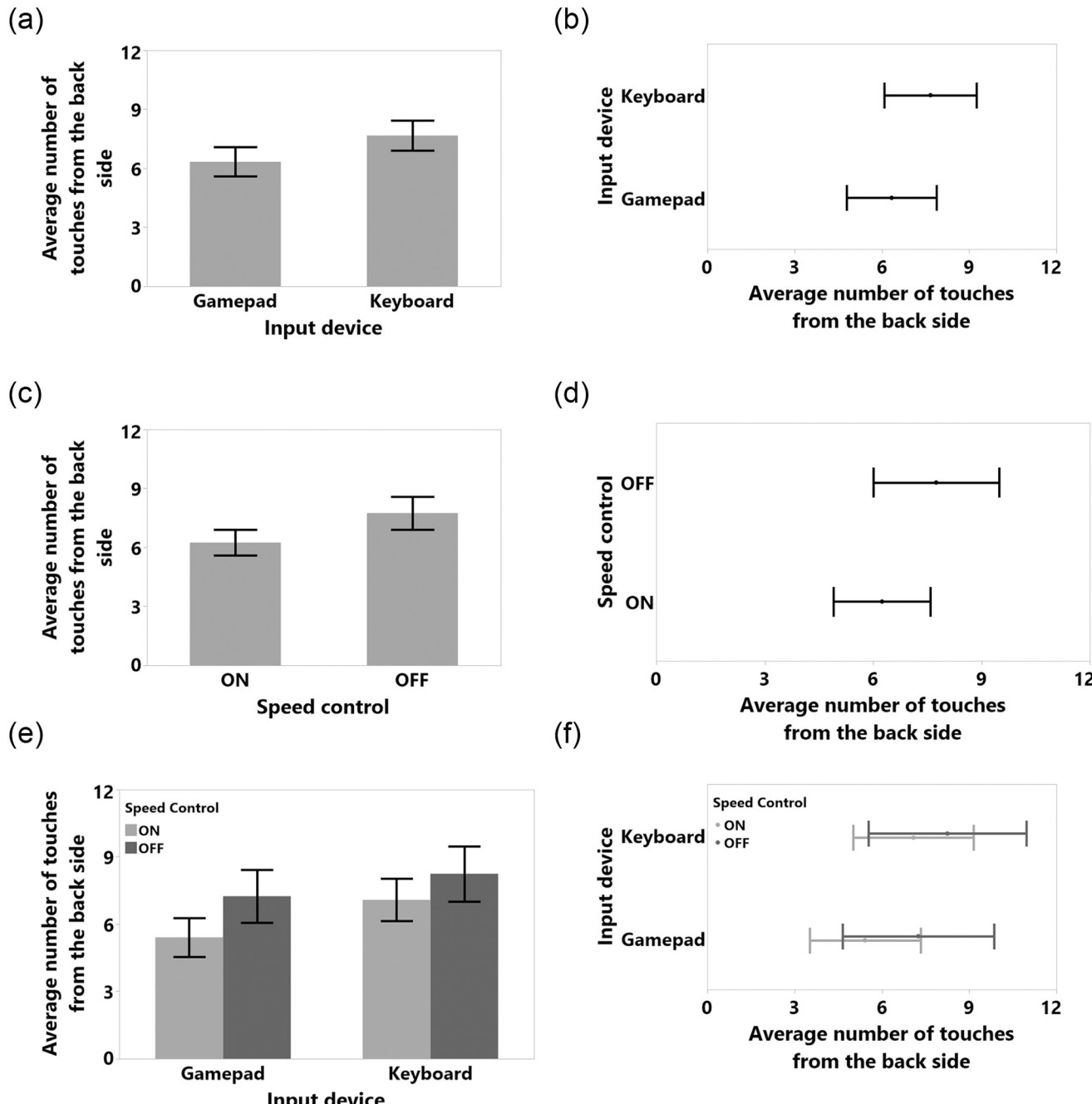

**Fig 9.** Means and standard error of means for number of touches from the back side for (a) input device, (c) speed control, (e) input device and speed control interaction. Means and 95% confidence intervals for number of touches from the back side for (b) input device, (d) speed control, (f) input device and speed control interaction.

avoid collisions with obstacles. The SC was not actively reducing the number of collision, but it was also acting as a warning mechanism by providing (indirect) visual feedback through the slowdown of the TR. According to the analysis of the time spent in individual milestones, subjects were significantly slower in milestones **M1-M2**, **M2-M3**, **M25-M26**, and **M26-M7**, see Fig 12(a). Interestingly, they also hit obstacles less often in these same milestones with the

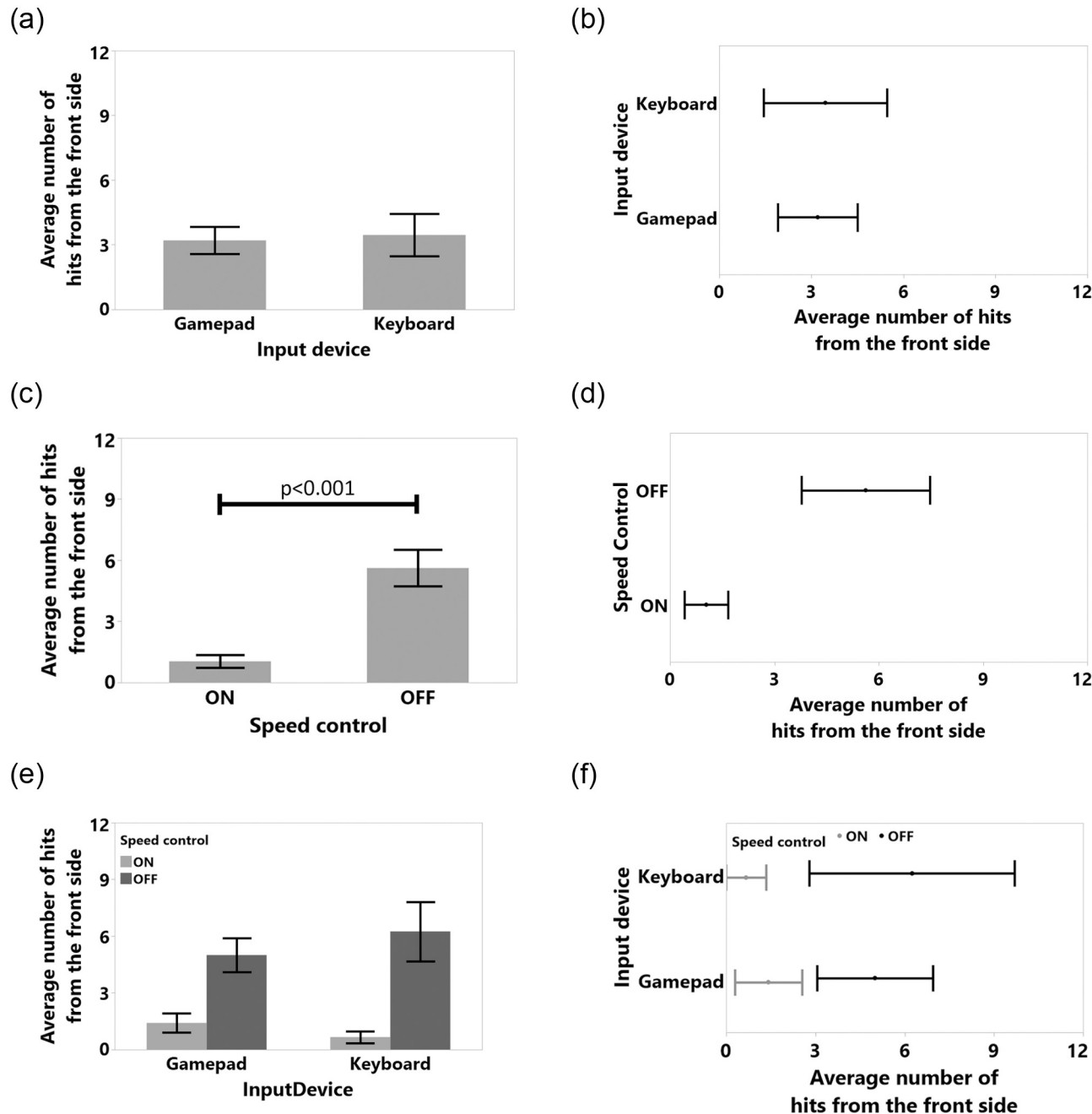

**Fig 10.** Average number of hits from the front side and standard error of means for (a) input device, (c) speed control, (e) input device and speed control interaction. Means and %95 confidence intervals for number of hits from the front side for (b) input device, (d) speed control, (f) input device and speed control interaction.

front side as visible in Fig 12(b), except for **M2-M3**. In **M2-M3**, SC decreased the number of hits, but we were not able to measure an effect. The milestones **M1-M2**, **M2-M3**, **M25-M26**, and **M26-M7** belong to segments where the path is strongly curved, as shown in Figs 4 and 5. The detailed milestone analysis results show that SC helped subjects to avoid hitting obstacles in tight, curved path segments, which took longer to navigate. However, in other milestones, such as **M16-M17** and **M0-M1** in Fig 12(b), **M32-M33** in Fig 13(b) and 13(c), and **M16-M17**

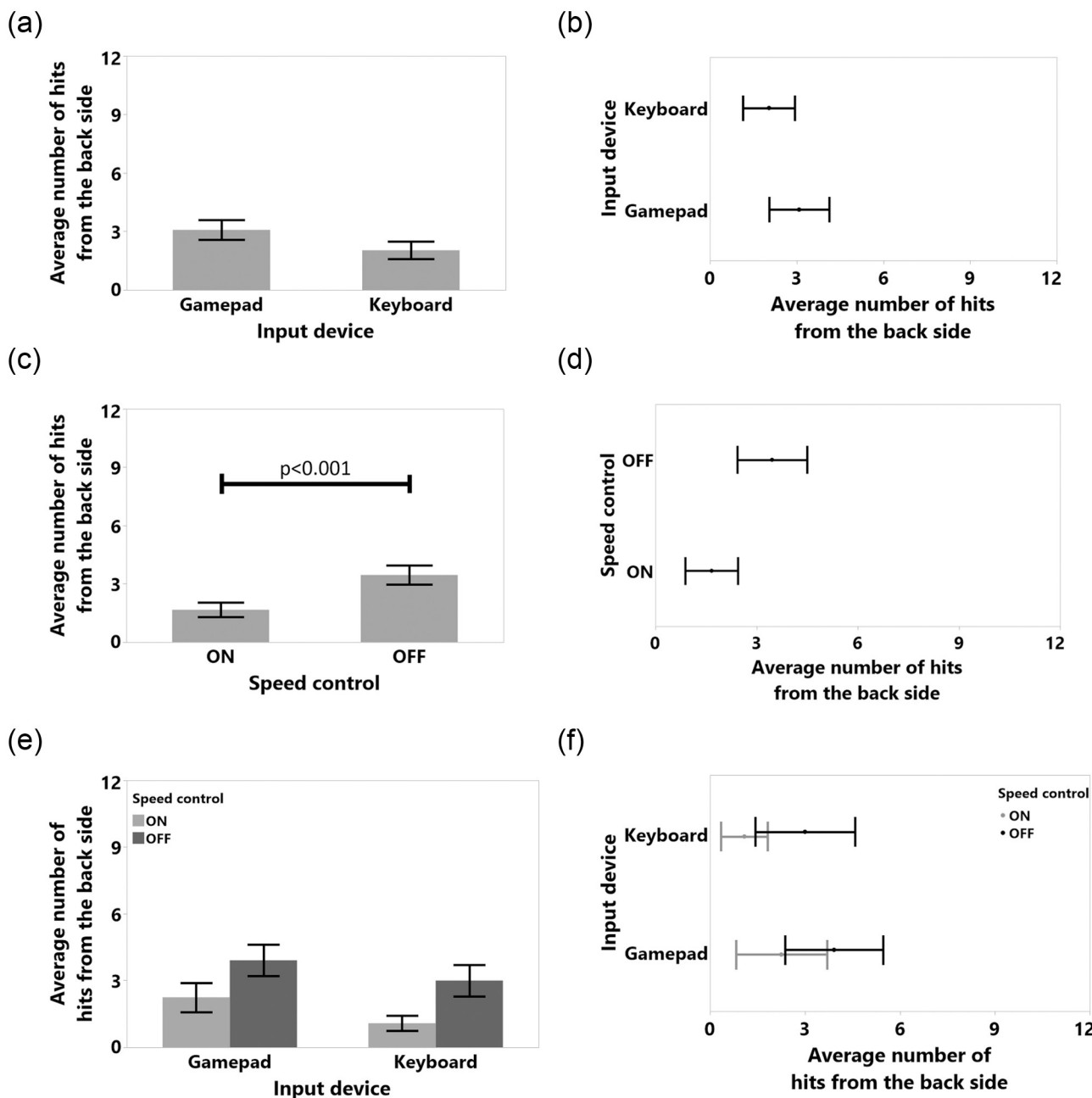

**Fig 11.** Means and standard error of means for number of hits from the front side for (a) input device, (c) speed control, (e) input device and speed control interaction. Means and %95 confidence intervals for number of hits from the front side for (b) input device, (d) speed control, (f) input device and speed control interaction.

in Fig 13(d), SC decreased the number of collision while not increasing the navigation time. The remainder of the results for the milestone analysis can be found in Fig 13.

**4.5.5 Subjective measurements.** At the end of the experiment, we asked participants to fill a short questionnaire about their experience, thoughts, and insights. We asked participants as to which driving method was preferred, with or without SC. All participants preferred SC over the condition without SC. They commented on the SC in various ways, including that

(a)

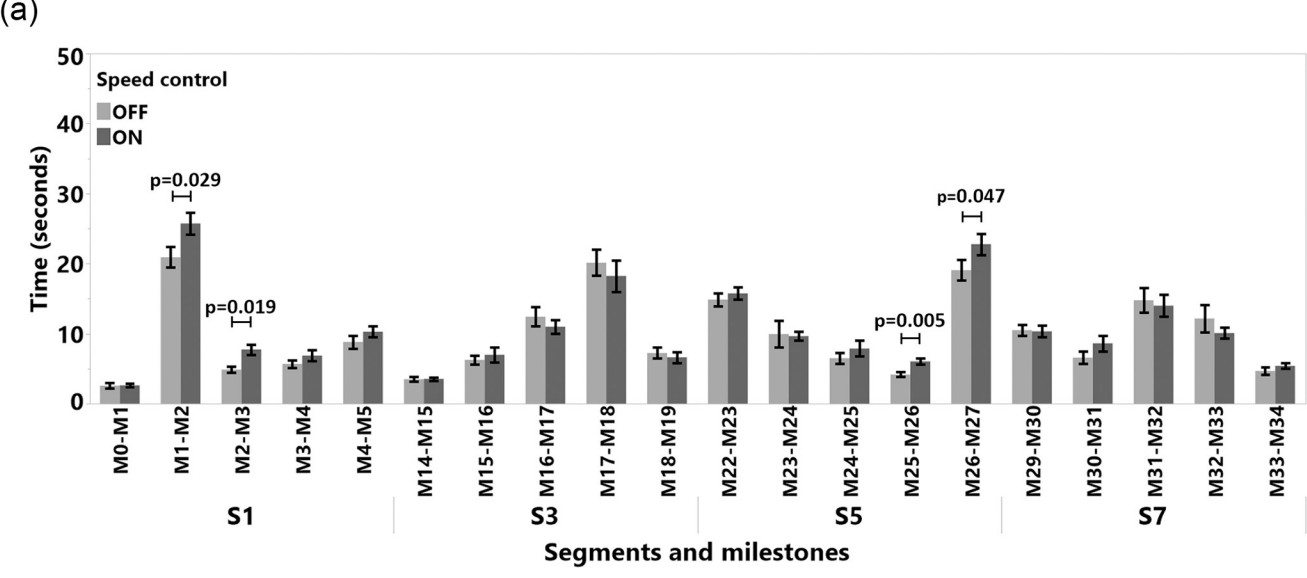

(b)

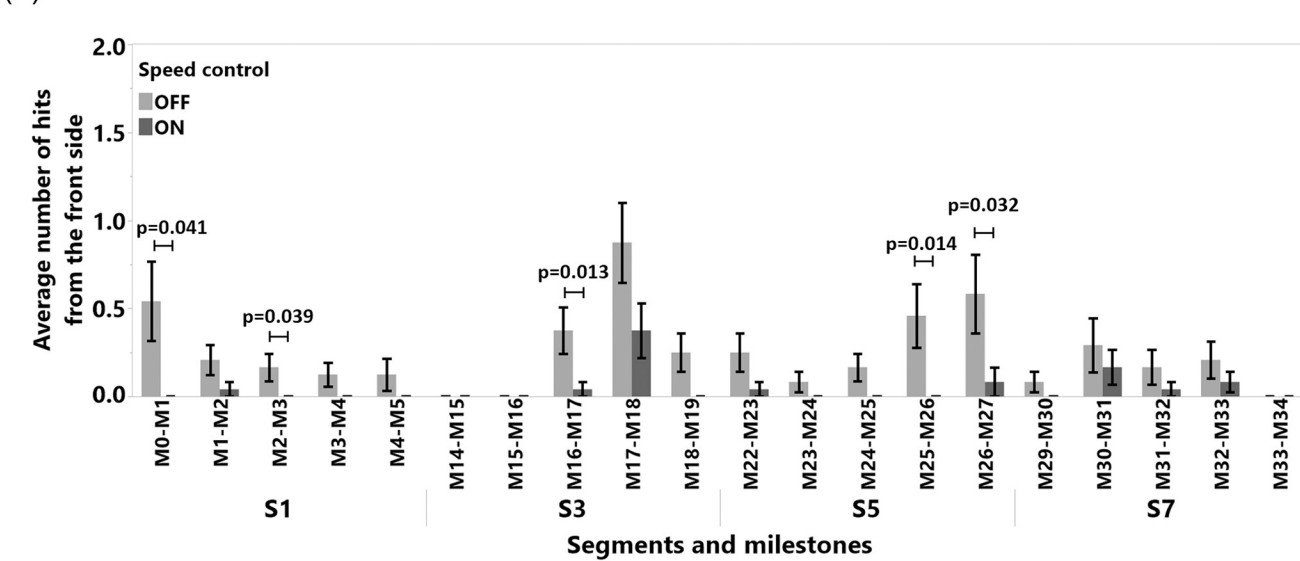

**Fig 12. Detailed milestone analysis for (a) time and (b) average number of hits from the front side.** Only path segments with significant differences in at least one of the measures are shown.

"they felt more safe", "less confusing", "easy to use" and "gave more time to control the device". We also used a 7-point Likert scale to evaluate user perceptions for the SC algorithm and input methods and analyzed the results to investigate subjective measures. None of the participants thought that it was difficult to drive the TR (1-easy, 7-difficult, Mean (M) = 2.42, Standard Deviation (SD) = 0.89). Only one participant reported that it was "somewhat difficult" to use the gamepad and the rest thought it was easy (1-easy, 7-difficult, M = 2.92, SD = 1.14). Only two participants thought that SC was "somewhat unlikely" to have improved their performance in terms of time (1-very likely, 7-very unlikely, M = 2.67, SD = 1.04) and none of them thought that SC worsened their performance in terms of number of collisions (1-very likely,

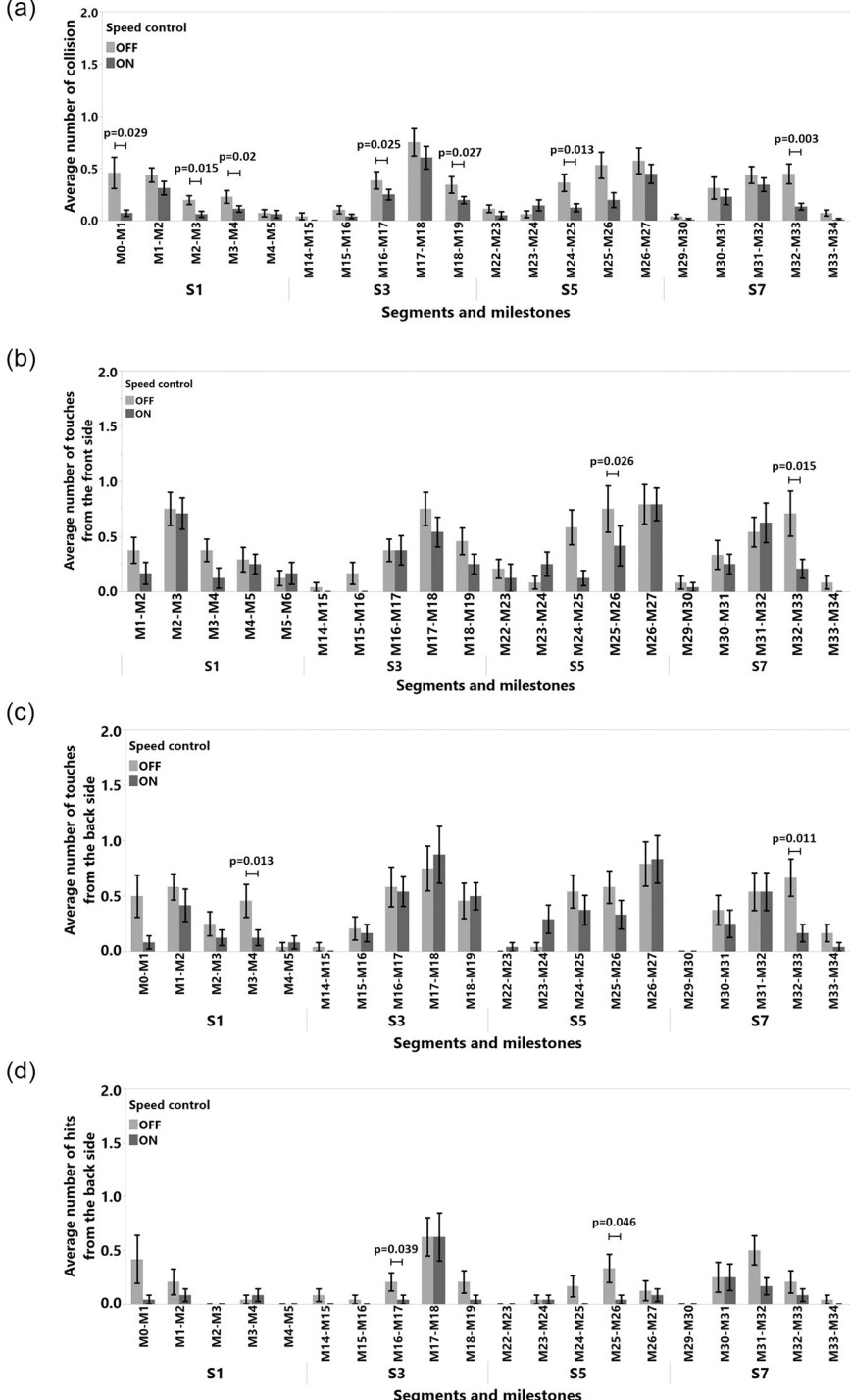

**Fig 13. Detailed milestone analysis for average number of (a) collisions, (b) front touch, (c) back touch and (d) back hit.** Detailed time and front hit analysis per milestone is shown in Fig 12.

7-very unlikely, M = 2.58, SD = 1.55). Half of the participants preferred the keyboard and the other half the gamepad. Participants also reported that they only felt moderate physical and mental fatigue after the experiment (1-I feel very rested, 7-I feel very tired, for physical and mental fatigue of M = 3.25, SD = 1.09 and M = 3, SD = 1, respectively).

### 4.6 Discussion of user study 1

Study 1 results suggest that adding automatic SC reduced the overall number of collisions when driving a TR, irrespective of whether the TR was controlled by gamepad or keyboard. These results also support our hypothesis, **H1**, that distance-based SC improves TR navigation behaviour in dense environments. The reduced number of collision also matches findings of previous TR studies. Yet, unlike previous work, SC did not decrease the task execution time [16–18, 40].

In the detailed analysis of segments and milestones, we observed that touches and hits occur less frequently when the SC is enabled. **M3-M4** and **M25-M26** involve a curved path that requires slow movements to traverse without hitting objects. This part of the maze also forces users to adapt to the speed-accuracy trade-off. Because the TR slowed down with SC, participants did not hit objects as frequently and thus got less upset when hitting objects (**H1**) or spent less time correcting their driving (**H2**), which might explain the lack of a time difference between the SC conditions. It also supports **H2** since the number of collision decreased with SC and users preferred the SC condition. Yet, we did not observe any speed-accuracy trade-off for individuals, see Fig 14.

Note, however, that automatic SC did not improve task completion time, even though all participants stated that they preferred driving the TR with added SC, mentioning amongst other factors the increased safety and ease of use. We speculate that this could be a positive side effect of hitting fewer objects in the maze. Instead of getting faster, subjects spent their time to carefully steer the TR through the maze so as not to hit objects. Since the task instructions mentioned both not hitting the objects and to finish the course as fast as possible, they perceived this as an positive outcome. Moreover, previous work showed that using SC, i.e.,

(a)

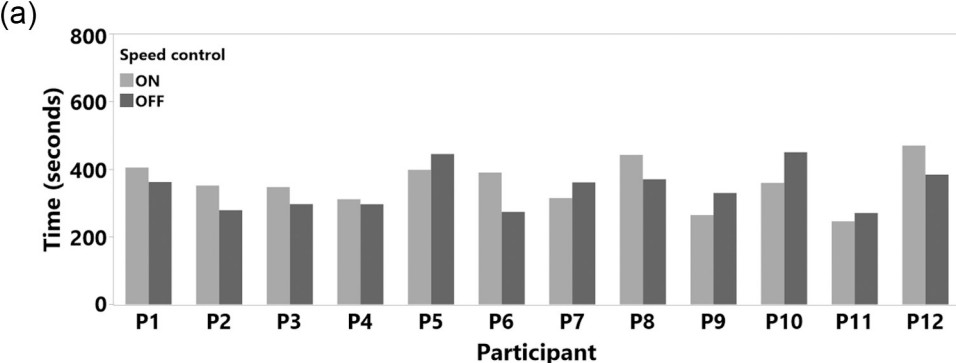

(b)

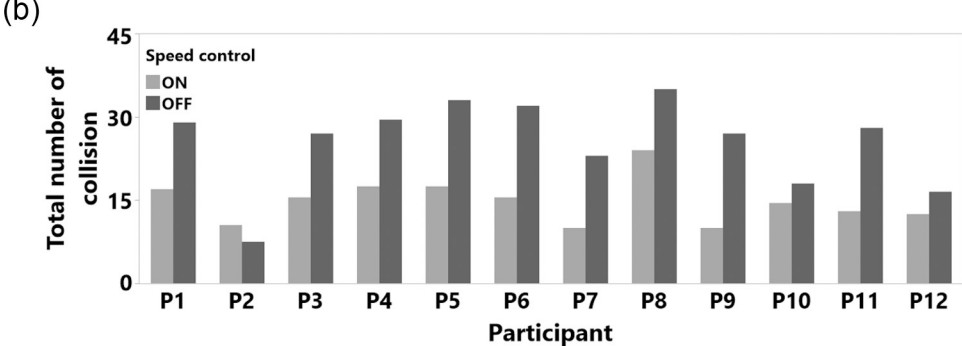

**Fig 14. Study 1 participant results (a) total task completion time and (b) total number of collision.**

reducing the speed of the TR, also increased task completion time [17, 18]. Yet, the SC algorithm used in this work did not significantly increase the task execution time, which we see as a positive indication.

The hits and touches that occurred at the front of the TR significantly decreased when the SC is enabled. Moreover, even though we did not collect and use the distance data from the back part of the TR, the number of hits on objects in the obstacle course with the back part of the TR significantly decreased. While we did not see such a difference for the touches at the front of the TR, a 360° distance range sensor with a higher data rate could increase reduce the number of touches with the front of the device, relative to what was observed with the limited-range ultrasound distance sensors we had at our disposal.

Through our interviews and questionnaire, we also learned that all the subjects preferred SC while they navigate in a static dense environment. We believe that continuously steering the TR in this static dense environment did require constant mental effort for navigation. As explained above, the SC algorithm helped subjects to reduce the number of collision by reducing the speed of the device automatically, which supports previous findings [17, 18]. Thus, participants had to worry less about hitting objects, which is the likely reason for all subjects preferring SC in Study 1.

Within this study, we analyzed how user behaviours changed with a SC algorithm and demonstrated that participants collide less with SC in static dense environment, as they can focus more on the challenges of navigating the robot along the path. We also demonstrated that it is possible to implement a SC method without having to alter the software or hardware of the TR itself.

## 5 User study 2

While the first study was designed to investigate user's maneuvering performance in dense static environments, it did not include any interaction with people, thus it is unclear how it might generalize to more typical conference-style situations and "live" remote environments. Moreover, participants navigated segments that were reasonably easy to memorize in a static dense environment, where they could benefit from SC. Towards environments that are more typical for TR operation, we designed a more ecologically valid conference-like environment as shown in Figs 15 and 16 using the same TR and software for Study 2. With this study, we aimed to investigate how user behaviours and their experience changes with a task requires social interaction with SC.

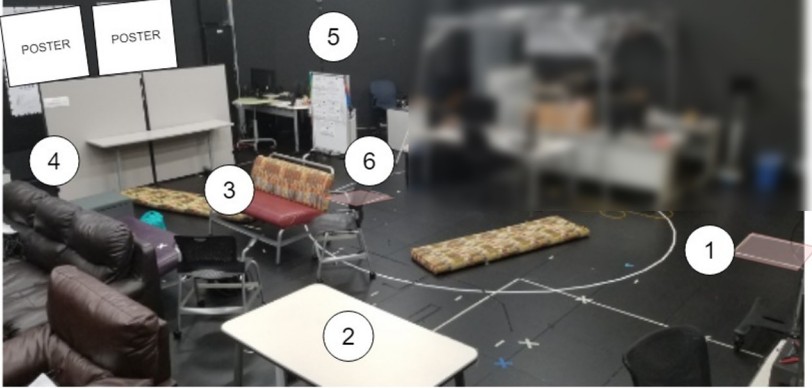

**Fig 15. View of the experimental setup for user study 2 for presentation.** Posters are blanked to preserve anonymity. The blurred area is associated with another research project irrelevant to this work.

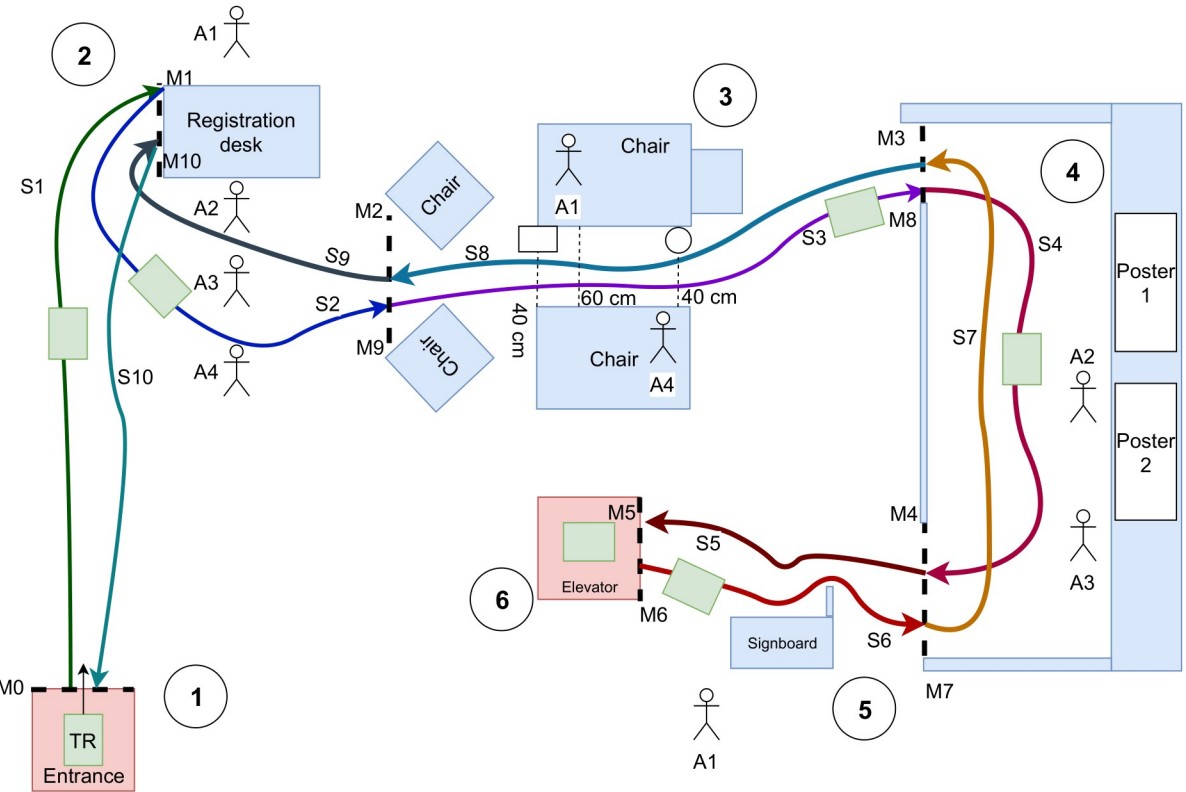

**Fig 16. Top down schematic view of the layout for study 2.** 1) Entrance and starting point of the experiment 2) registration desk 3) small corridor, 4) poster presentation hall 5) signboard 6) elevator).

## 5.1 Experimental setup

As the user study 1, we divided our experimental environment into eleven milestones that the user had to follow in the task. Since the experimental setup involves fewer sub-tasks, we only matched segments and milestones in this study. We also placed tape at the level of the sensors between chair/desk legs to ensure that the TR was able to reliably detect its distance to the chairs and desks. To provide a more realistic conference- or workspace-style setting, we used four actors/actresses in the environment while a participant was performing the tasks. These actors are shown as A1, A2, A3 and A4 in the Fig 16.

## 5.2 Participants

Twelve participants (9 female), with an average age of 27.6 with standard deviation of 4.8, participated our experiment. One participant was left-handed and all participants had never used a TR before.

## 5.3 Experimental design

Each of the 12 participants performed the experiment once. We divided the experiment into two sessions, without breaks. In the first session, participant started from **M0** and ended at **M5**, and in the second session they started from **M6** and ended at **M11**, as illustrated in Fig 16. Each of these two sessions started either with or without automatic SC, in counterbalanced order. Participants were informed if SC was turned on or off at the beginning of each session, i.e., at **M0** and **M6**. We again measured time and number of collisions in this study.

## 5.4 Procedure

This work involved a user study (Human Subject Research), conducted with approval of the Simon Fraser University Research Ethics Board (REB [2015s0283]). All participants signed informed consent forms and their data were analyzed anonymously.

After filling out the demographic questionnaire, participants were encouraged to get used to driving the TR until they felt comfortable. After that, the experimenter gave them a simplified diagram of the obstacle course, similar to Fig 16 but without the actor directions, milestones, and TR segments, and a sheet with tasks they had to follow. After participants read the instructions for the tasks on the sheet, the experimenter also verbally explained what participants had to do. In the first study, even though we were not able to identify any significant difference, gamepad slightly decreased task execution time and the number of collisions compared to the keyboard. Thus, we asked participants to use the gamepad in Study 2 to control the TR, as it allowed for continuous SC. Participants wore a headset with a microphone to talk to actors/actresses in the scene. A webcam on top of the desktop screen was used to show the face of the participant to the remote persons (in this case our actors). In other words, all interaction between the participant and actors/actresses throughout the experiment was through the TR system, as in Fig 16.

In segment **S1**, participants started the experiment from the entrance (1) and went to the Registration Desk (2). There was a queue for people to register at the conference, and participants had pass to the left of them and come close to Actor A1, who placed a badge onto the TR. In **S2**, participants had to navigate the TR through the people in the registration queue in front of the registration desk. Actors were positioned 50 cm away from each other and instructed not to automatically let the TR to go between them, such that participants had to interact with people in the queue. In **S3**, the TR had to drive between two chairs placed at 45˚. One actor/actress (A1) sat next to a laptop bag positioned on the floor as an obstacle, while another one (A4) sat on the other chair, in of a green bag. When A4 was sitting, there was insufficient space for the TR to pass by, so participants had to interact with the actors/ actress to either move the bag or ask for A4 to move their feet. In **S4**, the TR had to enter a "poster presentation area" and participant had to find answers for specific questions about the poster content. As a first task, participant had to find the correct poster, which required the participant to talk with the actors/actress (either A3 or A4). After finding the correct poster, participants had to respond to three questions: to count the number of figures in the results section, to read and write down a part of the sentence, and select a figure that stood out in terms of formatting. After **S4**, participants have to find the signboard in **S5** and find directions to five different locations. For these signs, we used names from a language unfamiliar to participants, which uses Latin letters (Hawaiian). Thus, participants had to ask for places with unfamiliar names. Three of these locations were indicated on the signboard and participants had to ask the student volunteer actor (A1) next to the signboard for directions to two of them. After recording the answers to the questions, participants drove the TR to the square which was designated as an "elevator", see Fig 16. At the end of **S5**, we finished data collection for the first SC condition. Without any breaks, we continued the experiment with the second SC condition. Before starting the experiment, subjects were informed if the SC algorithm was turned on or off.

When the participants started at **S6**, they first had to stop by the signboard and again respond to five questions on the task sheet in front of them, for a different set of destinations. They had to, again, fill the questionnaire and ask for directions to rooms and places in the Hawaii Convention Center. Again, two locations required communicating with the student volunteer next to signboard. After the signboard, participants went back to the poster

presentation area in **S7** and had to answer questions about the second poster. These question forced participants to interact with actor/actress A4. Participants had to write down the author names of the poster, which were not visible to the camera of the TR. They also had to navigate closer to the poster to see small details in the poster, such as the number of colored points in a graph and to count the number of experimental conditions. After **S7**, participants continued to **S8** and had to traverse between the same group of chairs as in **S3**, but in the opposite direction. Similarly, participants had to go through the line of people in segment **S9**. Again, none of the actors/actresses allowed the TR to pass through the line on their first attempt, such that participants were forced to interact with the actors in the line. At the end of **S9**, TR were asked to get close to A1's position such that A1 could retrieve the badge from the TR. At the end, participants drove back to the starting point, **S10** and finished the experiment.

In the two sessions of the experiment, i.e. session 1 starting from **M0** to **M5** and session 2 between **M6** and **M11**, we kept tasks very similar to each other. In other words, these sessions were symmetrical with small changes, i.e., in **M0**, participant had to take the badge and in **M11** gave the badge back to the actor. To counterbalance the conditions in our experiment, half of the participants started the experiment with the SC condition and the other half without automatic SC.

After the user study, participants filled a questionnaire about their preferences and insights. These questions were similar to Study 1, except that we omitted the questions around input devices.

We again used the video recordings of the Beam GUI to collect time data for each milestone indicated in Fig 16. We also counted the number of collisions in the video recordings.

## 5.5 Results

The $2_{SC}$ Speed Control data was analyzed using repeated measures (RM) ANOVAs for the independent variables, with $\alpha = 0.05$ in SPSS 24.

**5.5.1 Navigation behavioral data.** Completion time was normal after a logarithmic transformation (Shapiro-Wilk test result was $W(24) = 0.982$, n.s., Skewness = 0.566, Kurtosis = -0.422). The RM ANOVA results showed no significant main effects of SC ($F(1,11) = 2.492$, p = 0.143, $\eta^2 = 0.185$; with SC M = 325.5 seconds, SD = 84.2, SEM = 24.3, 95% CI [271.95, 379.04] and without SC M = 372.8 seconds, SD = 106.5, SEM = 30.7, 95% CI [305.01, 440.04]. Similarly, the number of collisions was not affected by the SC ($F(1,11) = 0.647$, p = 0.438, $\eta^2 = 0.056$). This might be related to the overall low number of collisions in study 2 (with SC M = 0.333, SD = 0.49, SEM = 0.14, 95% CI [0.02 0.64] and without SC M = 0.5, SD = 0.67, SEM = 0.19, 95% CI [0.07, 0.92]), likely a consequence of the more open areas compared to the environment in Study 1. Since we observed only a total of 10 collisions in this whole study, we only assessed the average number of collisions and did not analyze these hits in detail. We also did not see any clear evidence for a speed-accuracy trade-off in our results, as illustrated in Fig 17. The RM ANOVA results and statistical measures for Study 2 are shown in Table 3 and Fig 18. The detailed milestone analysis is shown in Fig 19. However, even though results were not significantly different, SC reduced the task completion time and number of hits, as shown in Fig 18.

**5.5.2 Subjective measurements.** In the post-experimental questionnaire, participants were asked the preferred driving method, with or without SC. After that, they were asked 7-point Likert scale to evaluate user perceptions for the SC algorithm.

5 participant preferred driving with SC, the other 7 participant preferred driving without SC. Participants who preferred SC commented such as "[speed control] prevents me from hitting the obstacles", "[the TR] was more controlled, and it felt comfortable", "The method with

(a)
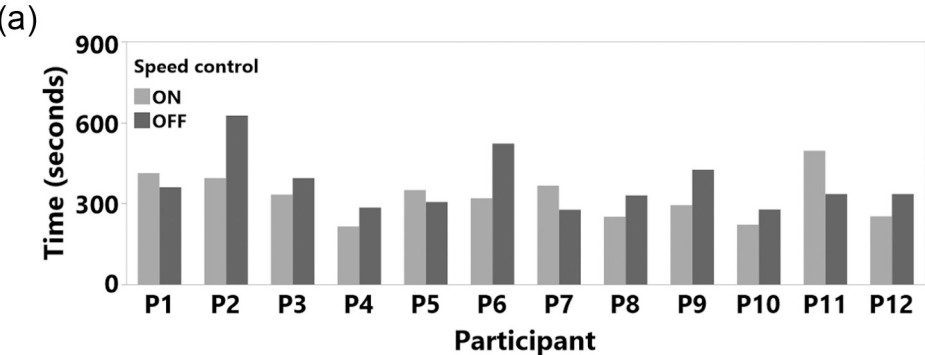

(b)
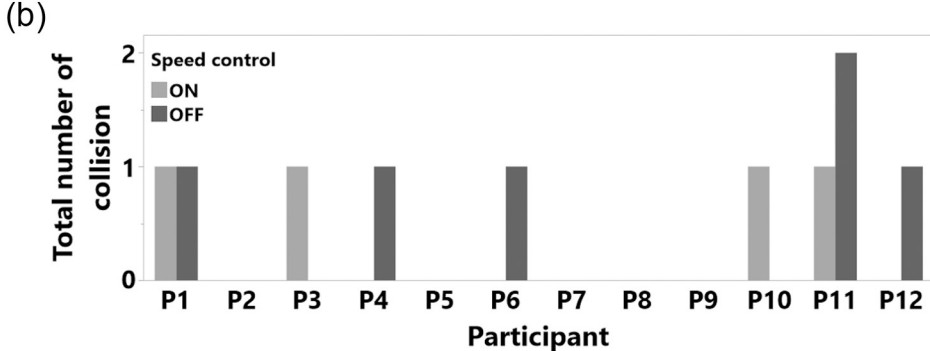

**Fig 17. Study 2 participant results (a) total task completion time and (b) total number of collision.**

speed control seemed to give more agency over the control of the robot. Without the speed-control method I found it more difficult to move around the space without hitting objects. The speed-control method also had smoother operation compared to the without speed control method.", "In my opinion it helps me have better control on my driving" or "it seemed smoother with speed control."

None of the twelve participants thought that it was difficult to drive the TR (1-easy, 7-difficult, the average result was 2.8). All twelve participants thought SC improved their performance in terms of task completion time (1-very likely, 7-very unlikely, M = 2.6, SD = 0.84) and none of them thought that SC worsened their performance in terms of the number of collisions (1-very likely, 7-very unlikely, M = 1.2, SD = 1.4). Participants also reported that they only felt moderate physical and mental fatigue after the experiment (1-I feel very rested, 7-I feel very tired, averages for physical and mental fatigue of M = 4, SD = 0.63 and M = 4.4, SD = 0.49 respectively).

Participants who did not prefer SC commented on their preference as follows: "I feel like the SC was going against my intention of wanting to get a closer look at something", "I felt more free to move. I didn't feel restricted", "I feel free to stop, move at my own will and gives me a more relaxed experience", "it felt more fluid", "less controls to worry about, especially

**Table 3. Study 2 ANOVA results and statistical measures.**

| Dependent Variable | Statistical Analysis | Speed Control ON | Speed Control OFF |
|---|---|---|---|
| Task Completion Time | $F_{(1,11)} = 2.492$, $p = 0.143$ $\eta^2 = 0.185$ | M = 325.5, SD = 84.2, SEM = 30.7, 95% CI [271.95, 379.04] | M = 372.8, SD = 106.5, SEM = 30.7, 95% CI [305.01, 440.04] |
| Total Number of Collisions | $F_{(1,11)} = 0.647$, $p = 0.438$ $\eta^2 = 0.056$ | M = 0.333, SD = 0.49, SEM = 0.14, 95% CI [0.02 0.64] | M = 0.5, SD = 0.67, SEM = 0.19, 95% CI [0.07, 0.92] |

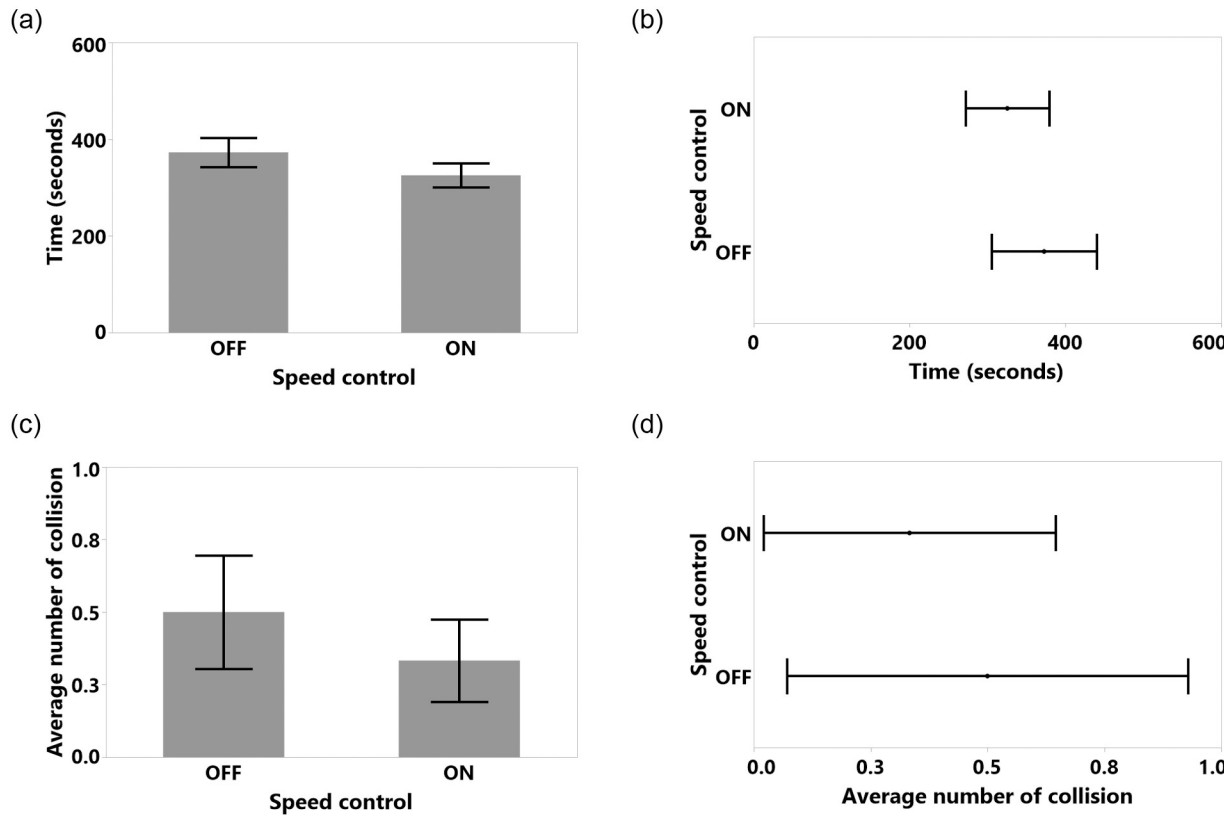

**Fig 18.** Study 2 SC task completion time (a) means and standard error of means, and (b) means and confidence intervals. Study 2 SC average number of collision for (c) means and standard error of means, and (d) means and confidence intervals.

when coming into this task with little experience using a game controller", "[without SC] was faster, get things done faster. I could steer around people faster in conference hall" or "made me think I had stumbled whereas I might just have been too close. This raises more question about personal space, but the lack of speed-control felt more organic and also provided an opportunity to learn how far I could go". None of the participants who preferred the condition without speed control thought that it was difficult to drive the TR (1-easy, 7-difficult, M = 2.28, SD = 0.89). Only a participant thought that SC was "somewhat unlikely" to have improved their performance in terms of task completion time (1-very likely, 7-very unlikely, M = 4.42, SD = 1.17) and none of them thought that SC worsened their performance in terms of number of collisions (1-very likely, 7-very unlikely, M = 1.91, SD = 1.1). Participants also reported that they only felt moderate physical and mental fatigue after the experiment (1-I feel very rested, 7-I feel very tired, for physical and mental fatigue of M = 3.28, SD = 1.33 and M = 3.71, SD = 1.56 respectively).

## 5.6 Study 2 discussion

In the second study, we investigated SC method in a conference-like environment where participant had to communicate with other people within a less dense environment compared to first study, while an automatic SC algorithm could be in effect.

Even though the TR slowed down with SC (though not significantly so) and SC slightly decreased the task completion time and the number of collisions, there were no significant differences on task completion time nor the number of collisions. Overall, study 2 did not show a

(a)

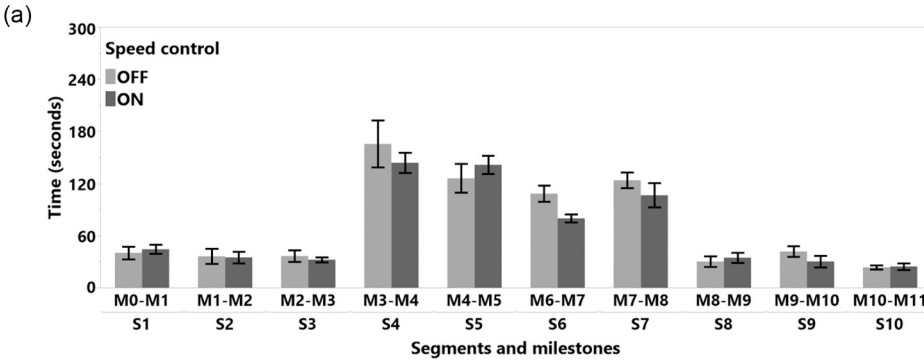

(b)

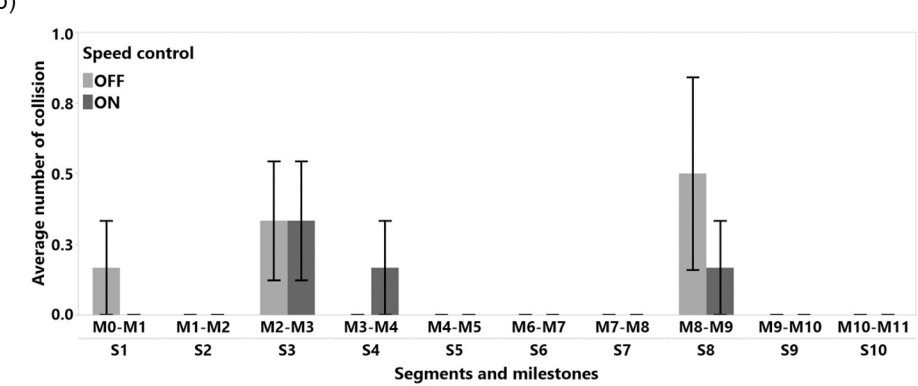

**Fig 19. Study 2 detailed milestone analysis for (a) task completion time and (b) average number of collision.**

significant improvement in terms of TR navigation behaviour when distance-based automatic SC was added, even though the device speed was reduced and number of collision decreased, thus not supporting **H1**. However, since subjects were socially interacting with other people, there are too many variables that might have influenced the outcome. Moreover, the open areas in study 2 did not allow us to investigate **H2**.

The observed split between participants who preferred to drive with or without SC suggests that people have different driving preferences [47]. Participants for whom full control, speed, and agency over the TR is important did prefer to have no automatic SC or at least the option to switch it off when desired, especially in low density spaces [16, 40].

In the first study, half of the participants had a valid driver's license. Three of them were driving once or twice a week and one of them was driving every day. Two participants with driver's licenses had been driving a car once or twice in six months. Also, half of the participants reported that they played car driving games regularly, the other half did not. In the second study, eleven participants had a valid driver's license. Three of them were were driving everyday, two of them were driving once or twice a week, two of them were driving once or twice a month, three of them were driving once or twice in six months, and one of them was driving once or twice a year. Two participants reported that they do play car driving games and the rest reported that they do not. Interestingly, all the participants who preferred to drive the TR without automatic SC had a valid drivers' license. Thus, one potential explanation of our results is that real-world car driving knowledge might affect participants desire for SC. However, there are other potential explanations, such as game car driving experience, cultural differences, or the influence of social interaction on driving, which could be investigated in future work.

One possible explanation of the results is the hardware design of the distance sensors. In our TR setup, we used sensors placed approximately 30 cm above the ground and they were measuring distance at this height. Objects (much) lower and higher than this height were not detected. For instance, while it was possible to measure the distance between the TR and the leg of an actor, the shoe of the actor was not detected. Yet, a (technically naive) end-user would likely expect the system to detect objects on the floor. When such obstacles were not detected, user had to manually control the TR (even though SC was enabled), which might have increased the frustration of the users and which could explain the outcome. We discuss the limitations of the current setup further below. We also speculate that different preferred social distances across cultures might have affected our results in here [48]. Our subjects had a culturally very diverse background, from all around the world. In some cultures, people prefer to get closer to each other to communicate. Thus, the SC algorithm could have negatively affected communication behaviours for different participants. We did not collect any data to assess this effect, but as this is also a potential explanation for our results it warrants further investigation. Another potential explanation of the results are environmental factors, such as noise, which can be found also during regular TR navigation. In the first study, subjects had to focus only on driving with visual feedback. In the second study, subjects had to interact with people through spoken communication. The noise in the environment or the incoming voice sound level might have had an effect on the users' preference to come closer to the actors, which might also have affected the results of our study in terms of preference for or against SC.

Beyond our actors answering questions, we did not observe other substantial social interaction between the participants and our actors, which is not that surprising considering that participants were typically trying to minimize their time. We were also unable to identify notable changes in terms of SC, i.e., slowing down near our actors. However, we did observe that one participant (S5) moved the TR around the line of waiting actors in S1, while the others went between the actors in the line. Among these participants, three of them (P7, P8 and P11) did not ask actors to move so that the TR could easily pass through the line of participants. In S3, one participant (P3) asked actor A4 to move their feet and one participant drove over the feet of A4 (P2), while the rest of the participants asked actors to move the bag in front of A4. In S8, the participant (P3) who asked A4 to move their feet in S3, went between the bag and A4 without informing A4. Lastly, in S9, two participants (P5 and P8) moved around the queue of waiting actors and two participants (P3 and P9) did not ask actors to move, but went through the line. The rest of the participants asked actors' permission to pass between them in the waiting line. We believe that these behavioural differences might be either due to varying cultural backgrounds or due to personality factors of the participants.

## 6 General discussion

In this work, we focused on changes in navigation user behaviors as an effect of using a speed control method. As a basis for this work, we implemented a SC method that uses the distance to objects in front of the TR to modulate the speed for dense environments, without modifications to the existing Beam+ TR nor its GUI. We then conducted a first user study in a static dense environment and demonstrated that such a SC method can improve user navigation behaviours in terms of fewer collisions, especially in tight sections with high curvatures in static dense environments. With the SC method participants felt safer to drive the TR and perceived an increased ease of use. This result also supports our hypothesis **H1**, i.e., that distance-based SC improves TR navigation behaviour. Even though we did not acquire data nor designed our system to reduce the number of collision at the back of the TR, we found that

collisions of the back of the robot decreased with the SC method. We believe that this result supports our **H2**, in that SC increased users' spatial presence in the maze. Since subjects were more able to pay attention to the environment, they also did not hit objects with the back of the TR in the maze. These two hypothesis also supports the findings of previous work on SC [40, 49].

In our second study we evaluated TR navigation behaviour with SC in an environment similar to an academic conference. In contrast to the first one, this study had larger open areas and subjects had to interact with real humans. We saw a trend towards a decrease in task completion time, but this was not significant, so our **H1** was not fully supported. Moreover, the large open areas in Study 2 did not allow us to further investigate our **H2** on spatial navigation.

**With Study 1, we tested a distance-based SC method in static dense environments and showed that it is beneficial for optimizing navigation behaviours in such settings**. All the subjects preferred SC in the first study, where they did not have to interact with people. Study 2 revealed approximately binary responses with respect to the sense of agency that users felt with respect to the two different SC methods. While about half of the participants preferred automatic SC because it allowed for smoother and safer navigation, the remainder preferred to have full control over the speed as this gave them more agency and allowed for faster travel even when close to obstacles. This result also supports the finding of previous studies on individual driving preferences with vehicles [47]. **While previous SC work highlighted the importance and the application of SC** [16, 40, 49], **our study revealed that always-on SC does not improve TR navigation behaviour across all environments, such as when users had to interact socially**. Still, the automatic SC method implemented in this work improved TR navigation behaviour in static dense environments for all users without increasing the task execution time compared to previous methods [17, 18].

Through our interviews, we learned that agency and control can play an important role in whether people prefer automatic SC or not. This outcome of Study 2 show similarity with other research in the field on TR navigation. Basu et al. [50] showed that participants' vehicle driving styles vary, yet all their participants preferred driving styles safer than to their own. This motivates us to believe that driving style affects user experience and the participants' desire for SC. Based on previous work [17], we speculate that adding automatic SC would reduce the user's perceived cognitive load when the TR comes closer to objects. In such situations the user does not have to divide their attention between the steering and the speed of the TR. Automatic SC frees up the mental resources of the user, as they do not have to focus on the speed of the TR and thus can focus on better navigation [17, 18]. Such limits to mental capacity have also been observed in other fields. For example, in pedestrian traffic research, it has been shown that there is a relation between walking speed, traffic density and the capacity of the environment [51]. When there are physical limitations in the environment, such as counter flow [52, 53] or obstacles, including stairs [54], human walking speed decreases in that particular environment to navigate more safely.

**Study 2 results highlight the importance of giving users control, so they can chose when they would like to have SC assistance or not, which presents guidance for the design of future TR systems with automation-on-demand** [51]. Other TR research on user driving skills also supports our conclusion; low-skill users benefit more from a TR assistance system designed to help users to avoid hitting obstacles than skilled users, i.e., the assistance system was not uniformly helpful to all user levels [18]. In the automotive domain, automatic SC assistance has been studied and results show that drivers prefer to take control of such systems in situations when the driver sees that the system is taking an action that is not desired [55, 56] Users could activate such features as desired or systems could be designed to detect the type of environment and adjust the degree of automation.

**The finding from subjective results also suggest that user preference needs to be taken into account when designing assistive features on TRs**—even though adding automatic SC showed a clear benefit by reducing collisions in Study 1, this did not generalize into an overall preference for it in Study 2 which involved social interaction and used a more naturalistic conference-style setting, which mixes tight passages with open areas, which were explored in previous work [16, 40]. Social interaction is an important aspect of TRs, but most work on assistance algorithms, such as [16–18, 40, 43, 57–61], did not test their algorithms in realistic environment with social interaction. **User preference on TR SC can change when users interact socially with other people in realistic environment**. It is also likely that because participants were in a more realistic environment in Study 2 and they interacted socially, a larger variety of factors, compared to Study 1, may have influenced their thought processes when driving. When in a natural environment, users of TRs must consider obstacles in addition to their desired interpersonal distances from others, which may change depending on the people who are around. Obstacles and people move around, so the dynamic nature of the environment could further affect people's preferences and the usability of the design. There are likely a host of ways that algorithms and systems could be designed beyond what we have discussed here. Regardless of the specific design solution, researchers and designers will need to consider such factors.

As stated above, we do not focus on the path-finding aspects for TR control. In our experiments, when the user lost their way, the experimenter helped participant to navigate back to the maze. This only occurred when the subject were in an open space, such as between **M5-M6**. In such instances, the TR could not hit or touch any objects. We also manually deleted all the data for intervals where the the user was trying to go back to the maze. This helped us to only focus our analysis on how collisions occurred in the dense environments. Further, we did not assess our SC results for certain steering conditions, such as in curved segments. These topics are already well-studied [62–65]. In this work, we focused our analysis on different segments and milestones to investigate if automatic SC shows any detrimental effects in various situations that could be found in *social gatherings*. Yet we were unable to identify any such negative effects.

Several measures of navigation performance had been previously used to assess the performance of machine-learning-based methods for social navigation behaviours in autonomous robots [66–72]. However, these studies focused on algorithms that automatically plan a socially acceptable path *without involvement of a human operator*. These approaches build on Proxemics theory [73] and use variables, such as distances between robot and person or path and trajectory length. In dense environments the path is typically so constrained, that path length (s) is not an appropriate measure to assess human-driven TR navigation (as path length is not a sufficiently sensitive measure in such environments). Also, in our study the tasks required TRs to get relatively close to actors and poster boards, much closer than what is typically considered desirable for autonomous navigation methods. Thus, we did not investigate navigation performance measures designed for autonomous robots in our studies. Navigation along narrow paths and in dense environments was also investigated within autonomous robotics, e.g., [74–76], but this research again did not involve human operators nor TRs.

**The results of our work also suggest that the evaluated SC algorithm can be used for static dense environments with narrow paths. When the user navigates into more open environments or interacts with people, the activation of the algorithm should be left to the user**. For instance, if there is a single object close to the side of the TR in a open space, or a single person in an open area, our algorithm would decrease the speed of the device automatically, which might not lead to the ideal driving experience. As future studies, additional features, different algorithms such as Conventional Neural Networks [77] or automatic way-

point approaches can be used to investigate user experience in such cases. Furthermore, as discussed in study 2, since this distance sensors used in this study work at a horizontal level, i.e., they only measure the distance at their installation height, objects out of view, i.e., sufficiently below or above this height, will not be detected, which might increase user frustration as they may not understand why the device did not "see" an obstacle.

## 7 Limitations

Even though our system was designed to investigate TR navigation in dense environments, our prototype still suffers from software and hardware limitations. Here, we acknowledge these limitation.

### 7.1 Sensor ring & distance measurements

In this study we used the six forward-facing ultrasound sensors attached to the TR based on existing hardware [38]. Yet, an increase in this number of sensors might also increase the accuracy of data acquisition. However, since we used an already existing setup, we were limited to use six sensors to achieve a sufficiently high update rate. Still, considering the size of the sensors and their view overlap, six seems to be a reasonable number. Also, the update rate of the system was dependent on several variables, such as the the speed of wireless connection and the processing rate of the computing hardware used in this experiment. While additional computing power might help, this will incur other issues, such as a need for more battery power. Although ultrasonic sensor rings have been used in previous TR studies for obstacle avoidance [59, 60] and navigation [58], the above-mentioned issue with not being able to detect the actors' feet also points out that a simple, horizontal ring of ultrasound sensors might not be entirely sufficient for collision avoidance nor automatic SC on a TR designed for conference-like environments.

To ensure that the distance sensors reliably detected all obstacles, we used duct tape to cover all open spaces, such as the space between the legs of chairs and open spaces between boxes in the first user study. We also taped all surfaces that were not reliably detected by the range sensors, such as metallic objects or small gaps in the "walls". However, a more precise system with a higher refresh rate, such as a LiDAR sensor or a millimeter cloud radar, could improve the rate and quality of the distance data acquisition, which then would also increase the performance of the SC algorithm. Yet, such systems are still prone to errors caused by reflective surfaces. Further, we acknowledge that is very challenging to measure the absolute position and/or speed of the TR reliably with the sensors we used, and thus we could not rely on these measures to improve the SC algorithm. Yet, the findings of our first study show that adding even our simple form of SC to a TR can improve the navigation behaviour and experience, especially in dense static environments, such as sections of a factory floor, where there are few people. In our second study, our results suggest that the SC algorithm could decrease task completion time and number of collisions, but we were unable to identify corresponding significant differences since subjects performed tasks with social interaction in a realistic environment. Overall, we believe that a SC method can be useful for TR applications.

### 7.2 Speed control algorithm

It is not feasible to replicate all potential social and static scenarios and study various UIs, GUIs, and SC algorithms. We designed a SC algorithm that controls the speed of the device specified through the GUI and assessed TR navigation behaviour in a static and a social environment to identify when automatic SC improves the user experience, so that we do not need to use additional feedback to the user. For instance, notifying the user about an upcoming

obstacle or making recommendations would not work well in the scenario for user Study 1, since all objects are close to the TR. This would lead to constant notifications, which would diminish the effect of the feedback and could be frustrating for participants. The movement speed of the TR is already visible through the optical flow in the camera feed: drivers can easily see how fast or slow the robot is moving by looking at the camera views. As such, we did not provide any additional feedback mechanism beyond the existing camera views of the remote environment. Considering such limitations, we implemented an unobtrusive TR SC algorithm and evaluated navigation behaviour in dense environments. Within the given software and hardware limitations, we were unable to identify or implement a notably better SC algorithm. Given that we have shown the benefits of a SC algorithm with Study 1, we consider a detailed comparison of the efficiency of different SC algorithms in various environments with different UIs to be out of the scope of this manuscript.

### 7.3 Telepresence robot

The TR we used in this work is designed for indoor environments, which is the reason why we limited it to a human's walking speed and designed the SC algorithm correspondingly. This also means that the outcomes of this work might need to be further investigated for outdoor scenarios, including, e.g., urban search and rescue robotic applications. While the top speed for our TR is in the range of speeds used in previous studies, such as [45, 53, 78, 79], we acknowledge that individual walking speed can vary with age, gender, and weight [79, 80] and we did not account for this variation in individual speeds in our study.

Similarly, the TR we used is a commercially available teleconference product, which was used in previous work [1, 8, 9]. Other TR systems with additional features, such as automatically adjusting to the user's head height [81], or with stereo displays with higher quality video transmission rate [82] exist. Moreover, unlike other telepresence work [43, 57, 61], we controlled the speed of the TR only through the user interface of the system, but not directly in the TR. In contrast, previous work that had full access to all components of the TR and where the feedback loop was running on a microcontroller, was able to use PID controllers [43, 57]. We believe that our system might perform even better with such a hardware platform.

### 7.4 Participants

We also collected data for 12 subjects in both studies. Although we used only a limited number of participants, we found clear differences, as also illustrated through overall high effect sizes (0.63 to 0.77). Even though the participants were different in both studies, a larger sample size could increase the depth of insights on the SC method in TR, especially for user study 2.

We also collected data from university students who had no previous experience with navigating a TR. Even though subjects were allowed to practice both with and without SC until they felt confident, a longer learning period or an expert user might experience different results [18]. Also, we had predominantly female participants in our studies, which is not representative of all target contexts.

### 7.5 Experimental environment

In both studies actors and obstacles were fixed. In a dynamic environment, such as a conference or a bustling workplace, people move around, also when they interact with the TR. Since it is difficult to replicate a real environment that constantly changes while maintaining the repeatability of the experiment, we decided to focus on static environments and obstacles. In a conference or work environment, the audio noise level would also be much higher than in our experiments.

### 7.6 GUI

In study 1, subjects did not need to interact socially, and pilot participants preferred a minimized forward-facing camera view. Thus, we increased the size of the downward camera view to its' maximum size for this study. Since the Beam GUI shows both views in a single window and does not allow one to split the camera views into different windows, that enlargement meant that the forward view automatically became as small as possible (350 pixel x 260 pixel or 11 cm x 8 cm). Still, this size was sufficient to enable participants to see where they were driving in free space and it also helped them to pass under the ladder (part of the obstacle course). In the second study, we changed the size of the both views to allow participants to experience reasonable views for both navigation in the environment and social interactions. Thus we gave similar amount of space to both camera views, 19 x 14.5 cm or 620 pixel x 465 pixel for the forward-facing camera and 18.5 x 14 cm or 595 pixel x 445 pixel for the downward-facing one. This is also shown in Fig 2. Thus, while we were carefully set the size of the camera views, these sizes (and the much larger than normal forward-facing camera view) should be taken into account when considering our results. To avoid a potential confound, we did not allow participants to individually vary the size of the camera views. These dimensions clearly depend on the size of the computer screen, which is another limitation of our work.

## 8 Conclusion and future work

In this paper, we assessed an automatic speed control method designed for telepresence robot navigation in dense environments. Our results showed that speed control can significantly improve TR navigation behaviour during telepresence robot navigation in static dense environments. Moreover, we also identified that in environments that require social interaction, the benefits of automatic speed control are reduced. Our findings also suggest that when the user had to interact within a social environment, some users could feel limited by automatic speed control, but for others the use of a speed control algorithm decreased cognitive load and improved spatial presence. Thus, we suggest that users should have an on/off option for automatic speed control algorithms. Moreover, researchers should test their developed algorithms with tasks that require social interaction. Researchers and designers of navigation and driving systems for telepresence robots should consider the design features we presented and explore how user preference, driving experience, environment, and desires for certain interpersonal distances from others can be accommodated in such systems.

In the future, we are planing to extend our studies to more complicated environments, such as where people are walking or passing in front of the telepresence robot as well as outdoor environments, where spaces may have differing types of obstacles, or uneven terrain with additional metrics, such as the NASA TLX. Another environment to study is areas where more people are present than what we have already studied, with various environmental conditions, such as different levels of noise. Moreover, we want to further investigate how previous driving experience affects telepresence robot navigation with a speed control algorithm.

## Supporting information

**S1 Video. The video of the submission.**
(MP4)

## Author Contributions

**Conceptualization:** Anil Ufuk Batmaz, Ernst Kruijff, Bernhard E. Riecke, Carman Neustaedter, Wolfgang Stuerzlinger.

**Data curation:** Anil Ufuk Batmaz.

**Formal analysis:** Anil Ufuk Batmaz.

**Investigation:** Carman Neustaedter, Wolfgang Stuerzlinger.

**Methodology:** Anil Ufuk Batmaz, Bernhard E. Riecke, Carman Neustaedter, Wolfgang Stuerzlinger.

**Project administration:** Carman Neustaedter, Wolfgang Stuerzlinger.

**Resources:** Jens Maiero, Bernhard E. Riecke, Carman Neustaedter, Wolfgang Stuerzlinger.

**Software:** Anil Ufuk Batmaz, Wolfgang Stuerzlinger.

**Supervision:** Carman Neustaedter.

**Validation:** Anil Ufuk Batmaz, Wolfgang Stuerzlinger.

**Visualization:** Anil Ufuk Batmaz.

**Writing – original draft:** Anil Ufuk Batmaz, Bernhard E. Riecke, Carman Neustaedter, Wolfgang Stuerzlinger.

**Writing – review & editing:** Anil Ufuk Batmaz, Ernst Kruijff, Carman Neustaedter, Wolfgang Stuerzlinger.

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
