## [Decision Letter · Decision Letter 0]

11 Sep 2020

PONE-D-20-21011

The Effect of Distance-Based Speed Control on User Behaviours in Teleperesence Robot Navigation within Dense Conference-Like Environments

PLOS ONE

Dear Dr. Batmaz,

Thank you for submitting your manuscript to PLOS ONE. After careful consideration, we feel that it has merit but does not fully meet PLOS ONE’s publication criteria as it currently stands. Therefore, we invite you to submit a revised version of the manuscript that addresses the points raised during the review process.

Additionally, please note that reference 57 was published in: Proceedings of World Conference on Educational Multimedia, Hypermedia and Telecommunications 2008, pp. 1706-1715, Chesapeake, VA: AACE

We look forward to receiving your revised manuscript.

Kind regards,

Catalin Buiu

Academic Editor

PLOS ONE

Journal Requirements:

Reviewers' comments:

Reviewer's Responses to Questions

**Comments to the Author**

1. Is the manuscript technically sound, and do the data support the conclusions?

Reviewer #1: Yes

2. Has the statistical analysis been performed appropriately and rigorously? 

Reviewer #1: Yes

3. Have the authors made all data underlying the findings in their manuscript fully available?

Reviewer #1: Yes

4. Is the manuscript presented in an intelligible fashion and written in standard English?

Reviewer #1: Yes

5. Review Comments to the Author

Reviewer #1: Overall

This is a strong paper with a nice presentation of a device that might help people navigate a telepresence robot through complicated spaces like a conference. The device is well described and the limitations are explicit. It uses a nice pair of user studies. I think it should be accepted with revisions outlined here along with the copy-editing in the associated document.

Lines 17-22. I was puzzled why you chose the speed control to add to the TR, but then by the time I got to line 22, you say “Inspired by work on SC in Virtual Reality….” Maybe you can delete the “In the work reported here…” and just start with “Inspired by work…” and add some of the “In the work reported here…” ideas there.

Line 20: As far as I can tell, you didn’t study “social behaviours of users,” you studied their performance. You probably could have done something with a qualitative analysis of how they slowed near people more than objects and how close they came, but that’s a lot of detail. Unless you add that, I’d rather you delete “social behaviours.”

Line 46: Delete the parentheses and keep the “while…”. It’s important. This is the speed/accuracy tradeoff. I think you should examine that in your data, especially in Study 2. Did the participants who tried to go fast have more collisions? It might be that the SC is really helpful for people who try to go fast, but is not necessary for those who naturally go more slowly.

Line 55: Add “necessarily” after “not.” This is important. There is question about what your Study 2 actually means, including your own admission in “limitations.” So, hedge the strong claim here.

Line 128: I didn’t understand “semi-autonomously” here. I think after reading the paper I do, but anything to help the reader know how this is semi-autonomous would help.

Line 252: “…sent steering commands to the TR.” Does it actually turn the TR or does it merely slow it down? Those are very different, in my opinion.

Around Lines 318 – 346: I noted in the margin that “at what height are the sensors? would it detect a dining room chair?” Your discussion of this limitation and how you handled it with tape should be previewed here somewhere.

Lines 378 – 380: This is a particular problem for putting these devices into use. I think you should comment on this as another limitation in the Discussion. Are other more expensive detectors subject to this limit?

Line 407: People took varying amounts of time to get used to steering this, I imagine. Can you report anything about this?

Figure 6: Why isn’t this one in Supporting Information, like the other graphs. Also, this is the first one I encountered that was totally meaningless when printed in B/W. I imagine some people will want to do that. Can you choose colors that show up as different grays when printed in B/W?

Procedure: Nice simulation of a conference venue. I wish you could have broken the analysis up into actions near people, including what they said and what the person did. It would have been nice.

Line 669: “With a different SC algorithm” I thought it was on or off, not different. This is very confusing.

Line 687: Did individuals who tried to go faster have more collisions. See my comment above on speed/accuracy tradeoffs and when the SC would likely be most beneficial.

Line 808-10: “SC decreased the task completion time and number of collisions,…and we were not able to measure an effect for time…” What? Is this the issue I noted in Lines 738-9 in the smaller edits? You can’t say it decreased because it was not significant.

Lines 886 – 896: Is this paragraph relevant. It covers 10 references, which are long to begin with.

Line 1041: Last line of the paper says “driving with automatic speed control follows the steering law.” What is the “steering law?” It’s not mentioned in the paper, as far as I can tell, and I read it pretty carefully. Does this refer to something that got deleted?

Copyediting

I don’t like stacked noun modifiers, as they are hard on a reader’s working memory. Therefore, I suggest the title: “How Automatic Speed Control Based on Distance Affects User Behaviours in Telepresence Robot Navigation within Dense Conference-like Environments.”

Lines 6,7: I don’t see how “academic conferences” are different from “professional meetings.” And the phrase “home schooling” is not what the Newhart work is about. That’s where parents choose to school their child/ren at home because they want to teach a certain way, like based on a religion. I think “school for homebound children” is more accurate. The children and parents do not choose to be at home; the health of the child bars them from school.

Line 11: Insert “both” in front of “obstacles” so the reader knows what to do with that first “and.”

Line 34: Start a new paragraph with “Our work builds…” but that sentence is 5 lines long. Break it up somehow.

Line 52: Start a new paragraph with “These results suggest…”

Line 73: Odd transition to last sentence.

Line 76: TRs in schools are often dressed in a t-shirt, at least. And, the robot is only used by a single student.

Line 124: This line needs something like “and so they don’t apply”.

Line 131: “Rasperbery” is meant to be “Raspberry”. I thought originally this was misspelled on purpose, like many systems are, but later it is spelled like the fruit.

Line 143: Substitute “factors” for “reasons.” You already have “caused”

Line 233: Figure 2. At first, I didn’t know what the “user-camera view” but later realized this was the user’s face as it appears on the TR. Is there some other words that could convey that?

Line 274: Remove the space between “methods” and “,”.

Line 279: Delete the “s” from “uses”.

Line 295: Start a new paragraph with “Before starting…”

Line 450: Here you call it an Appendix (capital A) but later it’s called “Supporting Information”.

Line 599 – 603: Using the phrase “different SC method” when one condition is with it and other not really threw me, especially Line 600 which has the “SC2” in it.

Line 626: “Chair” needs an “s”.

Line 704: Is “standard” = “without speed control”?

Lines 738-9: What you’re saying is that the direction of the difference was what was expected but it wasn’t significant. Saying “Even though the TR slowed down with SC…” is a conclusion that is not warranted. With no difference, it would be that half the time. One test of whether it is likely to be different is to calculate with your actual variance how big an N you would have to have to make it significant. I suspect you have high variance so the number would be huge, and therefore just “no difference.”

Line 782: Need a space between the “.” And the “Another…”.

Line 899: Add “s” to “interact”.

6. PLOS authors have the option to publish the peer review history of their article (what does this mean?). If published, this will include your full peer review and any attached files.

Reviewer #1: **Yes: **Judith S Olson

---

## [Author Response · Author response to Decision Letter 0]

23 Oct 2020

Dear editor,

We hereby submit a revised version of our manuscript “How Automatic Speed Control Based on Distance Affects User Behaviours in Telepresence Robot Navigation within Dense Conference-like Environments”. We take this opportunity to thank all reviewers sincerely for their time and comments, which have helped us to improve our manuscript. We have addressed all reviewer comments, suggestions, and critical remarks in the revision. Here is a summary of the changes.

Introduction 

As suggested by the external reviewer, we re-arranged the introduction. We highlighted that we focus on the navigation behaviours of users. 

Speed-accuracy trade-off

Even though we found evidence for a speed-accuracy trade-off for individual segments and milestones in study 1, we were unable to observe this at a per-participant level in both studies, as there was no inverse relationship between speed and collisions. For study 1 and 2, we now show these results in Figure 18 and 19, respectively.

Previous work

We edited the sentence discussing the use of TR in schools as follows:

“Despite the benefits, TRs are subject to operational challenges, as robots are often shared asynchronously amongst users in work and conference settings; thus, no one individual owns the “embodiment” (the look and sound, e.g., [5,20,21]) and remote users are unable to customize the robot’s appearance unless physical items are attached by a user who is local to the robot [22].”

Discussion/justification

We edited the following paragraph, where we better highlight our paper’s contribution in comparison to related work, as follows: 

“Several measures of navigation performance had been previously used to assess the performance of machine-learning-based methods for social navigation behaviours in autonomous robots [66-72]. However, these studies focused on algorithms that automatically plan a socially acceptable path \\textit{without involvement of a human operator}. These approaches build on Proxemics theory [73] and use variables, such as distances between robot and person or path and trajectory length. In dense environments the path is typically so constrained that path length(s) is not an appropriate measure to assess human-driven TR navigation (as path length is not a sufficiently sensitive measure in such environments). Also, in our study the tasks required TRs to get relatively close to actors and poster boards, much closer than what is typically considered desirable for autonomous navigation methods. Thus, we did not investigate navigation performance measures designed for autonomous robots in our studies. Navigation along narrow paths and in dense environments was also investigated within autonomous robotics, e.g., [74-76], but this research again did not involve human operators nor TRs.”

We also now describe the limitations of our approach more clearly, such as that we had to compensate for the shortcomings of ultrasound sensors by taping the environment and that placing the sensor ring at 30cm above the ground made detecting feet harder.

We also further discuss how more expensive detectors might perform. LiDAR, for instance, is also affected by errors caused by reflective surfaces, such as reflective surfaces that can be found indoors, e.g., on cabinet doors. 

Clarity

Based on the external reviewer’s suggestion, we changed the title to “How Automatic Speed Control Based on Distance Affects User Behaviours in Telepresence Robot Navigation within Dense Conference-like Environments.” We removed our mention of the steering law mentioned. As identified by the external reviewer, we revised all text about insignificant results. We changed “supporting information” to the Appendix. We replaced the word ‘factors’ with ‘reasons’, and fixed Rasperbery to Raspberry. We also deleted sentences with odd transitions or which were confusing for the reader. We also clarified our terminology for speed control and how the second experiment was conducted. We changed the use of ‘semi-autonomous’ to read ‘while the user can navigate the TR’ and rewrote ‘…sent steering commands to the TR’ to ‘sent commands to the TR to modify its speed during steering’. 

Procedure

We also did not observe any strong differences in terms of learning to steer the TR across different conditions. We reported this effect as follows:

“Based on our observations we can report that participants took about 5 minutes to complete the training phase. Given that our participants had not driven TRs before, we did not notice any strong differences in terms of learning the different conditions.”

We also now mention our observations around the interaction between the participant and the actors through the TR.

“Beyond our actors answering questions, we did not observe other substantial social interaction between the participants and our actors, which is not that surprising considering that participants were typically trying to minimize their time. We were also unable to identify notable changes in terms of SC, i.e., slowing down near our actors. However, we did observe that one participant (S5) moved the TR around the line of waiting actors in S1, while the others went between the actors in the line. Among these participants, three of them (P7, P8 and P11) did not ask actors to move so that the TR could easily pass through the line of participants. In S3, one participant (P3) asked actor A4 to move their feet and one participant drove over the feet of A4 (P2), while the rest of the participants asked actors to move the bag in front of A4. In S8, the participant (P3) who asked A4 to move their feet in S3, went between the bag and A4 without informing A4. Lastly, in S9, two participants (P5 and P8) moved around the queue of waiting actors and two participants (P3 and P9) did not ask actors to move, but went through the line. The rest of the participants asked actors’ permission to pass between them in the waiting line. We believe that these behavioural differences might be either due to varying cultural backgrounds or due to personality factors of the participants.”

Figures

We changed the colours of the figures to use different gray levels, which will show up better when printed in greyscale. We also changed the user-view notation in Figure 2. The documentation on the Beam website calls the user-facing camera the “self-view”, so we changed our text to read “the participant's self-view captured through the webcam of the computer”.

Minor issues

We also copyedited the paper and fixed minor grammatical errors. Also, as suggested by the external reviewer, we broke long sentences into smaller ones, broke long paragraphs into smaller ones, and changed the paragraph endings based on the suggestion of the reviewer.

References

As identified by the editor, we fixed reference [57]. 

PlosOne Style

We used the Latex format provided by the official PlosOne Website, https://journals.plos.org/plosone/s/latex.

Data

As previously mentioned, our data will be available in an online repository after acceptance, at https://osf.io/anxv3 DOI: 10.17605/OSF.IO/ANXV3.

---

## [Editor Report · Decision Letter 1]

27 Oct 2020

How Automatic Speed Control Based on Distance Affects User Behaviours in Telepresence Robot Navigation within Dense Conference-like Environments

PONE-D-20-21011R1

Dear Dr. Batmaz,

We’re pleased to inform you that your manuscript has been judged scientifically suitable for publication and will be formally accepted for publication once it meets all outstanding technical requirements.

Kind regards,

Catalin Buiu

Academic Editor

PLOS ONE
---

## [Editor Report · Acceptance letter]

5 Nov 2020

PONE-D-20-21011R1 

How Automatic Speed Control Based on Distance Affects User Behaviours in Telepresence Robot Navigation within Dense Conference-like Environments 

Dear Dr. Batmaz:

I'm pleased to inform you that your manuscript has been deemed suitable for publication in PLOS ONE. Congratulations! Your manuscript is now with our production department. 

Kind regards, 

on behalf of

Prof. Catalin Buiu 

Academic Editor

PLOS ONE